# SOFT PREFERENCE OPTIMIZATION: ALIGNING LANGUAGE MODELS TO EXPERT DISTRIBUTIONS

## ABSTRACT

We propose Soft Preference Optimization (SPO), a method for aligning generative models, such as Large Language Models (LLMs), with human preferences, without the need for a reward model. SPO optimizes model outputs directly over a preference dataset through a natural loss function that integrates preference loss with a regularization term across the model's entire output distribution rather than limiting it to the preference dataset. Although SPO does not require the assumption of an existing underlying reward model, we demonstrate that, under the Bradley-Terry (BT) model assumption, it converges to a softmax of scaled rewards, with the distribution's "softness" adjustable via the softmax exponent, an algorithm parameter. We showcase SPO's methodology, its theoretical foundation, and its comparative advantages in simplicity and alignment precision.

## 1 INTRODUCTION

The alignment problem focuses on adjusting a generative model (e.g., Large Language Models (LLMs)) to align its outputs with human preferences and ethical standards or to tailor the model for specific tasks; and is especially important after supervised fine-tuning on datasets with mixed-quality samples. A widely embraced approach involves refining these models based on expert (i.e., human) preferences, typically expert-provided comparisons of pairs of model-generated outputs (Christiano et al., 2017). Given a *preference dataset* $\mathcal{D}$ and a pre-trained model $\pi_{\mathrm{ref}}$, preference alignment seeks to train a new model, $\pi_\theta$, whose outputs are better aligned with the preference in $\mathcal{D}$ (Radford et al., 2018; Ramachandran et al., 2016). A notable advancement in this field has been the application of Reinforcement Learning from Human Feedback (RLHF), which involves training a reward-model based of actions preferred by humans and then optimizing $\pi_\theta$ to maximize these learned rewards while ensuring closeness to the initial model behaviors (Ouyang et al., 2022). Despite the effectiveness of RLHF in addressing the alignment problem, RLHF involves a relatively complex pipeline, susceptible to propagation of reward-model's biases over to the policy optimization.

Recently, several studies have introduced methods for the direct optimization of preferences, including Direct Preference Optimization (DPO) among others (Rafailov et al., 2023; Amini et al., 2024; Chowdhury et al., 2024; Xu et al., 2024; Yin et al., 2024; Xu et al., 2023; Tunstall et al., 2023). These approaches eliminate the need for a separate reward model training phase, instead adjusting the model directly using preference data, and often outperform RLHF-based approaches. These reward-model-free methods enjoy advantages over RLHF-based approaches, such as simplified pipelines, reduced computational complexity, and avoidance of the bias transfer from the reward model to policy optimization. Indeed, the rationale for incorporating an additional component, the reward model, into a supervised learning context with a supervised dataset, is debatable.

In this work, we propose a simple and effective reward-model-free alignment method, termed *Soft Preference Optimization* (SPO). SPO seeks to align the model's *preference estimates* (detailed in Section 3) with expert preferences $\mathcal{D}$, through minimizing a loss function of the form

$$\mathrm{AlignmentLoss}(\pi_\theta, \pi_{\mathrm{ref}}, \mathcal{D}) = \mathrm{PreferenceLoss}(\pi_\theta, \mathcal{D}) + \mathrm{Regularizer}(\pi_\theta, \pi_{\mathrm{ref}}), \qquad (1)$$

where the Regularizer may be chosen as the KL divergence. We discuss natural choices for the model's preference estimates and the preference loss function in Sections 3 and 4.

Unlike RLHF and DPO, the development of SPO does not rely on assumptions regarding the existence of underlying rewards, such as the Bradley-Terry (BT) model (Bradley & Terry, 1952). Nevertheless,

we demonstrate that if the BT model is applicable and given an asymptotically large preference dataset, SPO is theoretically guaranteed to converge to a softmax of the rewards, which inspires the designation "*Soft* Preference Optimization". Unlike DPO, which tends toward a deterministic model even with an extremely large dataset if the regularization coefficient is nearly zero (Azar et al., 2023), SPO allows for the adjustment of the softmax's exponent through an input parameter, thereby offering flexibility in modulating the "softness" of the output distribution.

SPO has two main distinctions from its successor reward-model-free alignment methods. The first distinction involves the choice of a preference loss that aligns model's preference estimates with expert's preferences, resulting in a favorable fixed point as discussed in the previous paragraph. The other distinction of SPO with DPO and similar algorithms lies in the application of regularization. DPO restricts regularization to the preference dataset, which is counter-intuitive since the dataset already provides specific data points for the model to fit; thus, additional regularization within this limited scope is unnecessary. More critically, since the preference dataset represents a tiny subset of the potential outputs of the model, focusing regularization solely within this subset can lead to undesirable, extensive shift in the model's distribution outside of the dataset, resulting in a non-coherent behaviours. Acknowledging this limitation, SPO applies regularization across the entire output distribution of the model, not just within the confines of the preference dataset.

## 2 BACKGROUND

Consider a finite context (or query) space $\mathcal{X}$ and a finite action (or response) space $\mathcal{Y}$. For a given query $x \in \mathcal{X}$, a behavior policy (such as a pre-trained model) is employed to generate responses $y_1, y_2 \in \mathcal{Y}$. These responses are subsequently evaluated by expert raters (e.g., humans) to determine which of $y_1$ or $y_2$ constitutes a more appropriate response to the query $x$. We adopt the notation $y_1 \succ y_2$ to denote that $y_1$ is preferred over $y_2$ in a specific context. The true expert preferences are typically represented by a probability, $p^*(y_1 \succ y_2|x)$, reflecting the inherent randomness due to the variable nature of the experts, who may be a group of humans with slightly differing preferences. A preference dataset, $\mathcal{D}$, is compiled by collecting expert preferences for multiple $(x; y_1, y_2)$ tuples. In detail, $\mathcal{D}$ comprises tuples $(x; y_w, y_l)$, where $y_w \succ y_l$ indicates the preferred (winner) and less preferred (loser) responses based on expert evaluations.

**RLHF** comprises two main phases: reward modeling and reinforcement learning (RL) fine-tuning. The initial phase, reward modeling, operates under the assumption that there exist latent rewards $r(y|x)$ that form the basis of expert preferences. This phase aims to develop a model capable of closely approximating these underlying rewards. A widely accepted method for defining these latent rewards is through the Bradley-Terry (BT) model (Bradley & Terry, 1952), alongside the Plackett-Luce (PL) ranking models (Luce, 2005; Plackett, 1975). The BT model posits that the distribution of expert preferences, $p^*$, is characterized by

$$p^{\mathrm{BT}}(y_1 \succ y_2|x) \stackrel{\text{def}}{=} \sigma\big(r(y_1|x) - r(y_2|x)\big) = \frac{\exp\big(r(y_1|x)\big)}{\exp\big(r(y_1|x)\big) + \exp\big(r(y_2|x)\big)}, \qquad (2)$$

where $\sigma(\cdot)$ represents the sigmoid function. Subsequently, the reward model $r_\phi(y|x)$ is trained to minimize the negative log-likelihood loss, $-\mathbb{E}_{(x;y_w,y_l)\sim\mathcal{D}}\big[\sigma\big(r(y_w|x) - r(y_l|x)\big)\big]$. The PL model generalizes the BT model for data involving rankings, modeling the expert distribution as

$$p^{\mathrm{PL}}(y_1 \succ \cdots \succ y_n \,|\, x) \stackrel{\text{def}}{=} \prod_{k=1}^{n-1} \frac{\exp\big(r(y_k|x)\big)}{\sum_{i=k}^{n} \exp\big(r(y_i|x)\big)}, \qquad (3)$$

for all $(x; y_1, \ldots, y_n) \in \mathcal{X} \times \mathcal{Y}^n$.

The RL fine-tuning phase aims to train a model, $\pi_\theta$, to maximize a loss function of the form

$$\mathcal{L}_{\mathrm{RLHF}}\big(\pi_\theta, \pi_{\mathrm{ref}}, r_\phi\big) = -\mathbb{E}_{x\sim\mathcal{D}, y\sim\pi_\theta(\cdot|x)}\big[r_\phi(y|x)\big] + \beta\mathcal{D}_{\mathrm{KL}}\big(\pi_\theta \,\|\, \pi_{\mathrm{ref}}\big), \qquad (4)$$

where $\beta$ is a non-negative constant, $r_\phi$ is the trained reward function, and $\pi_{\mathrm{ref}}$ is a reference policy often acquired through supervised fine-tuning on high-quality data and is typically identical to the behavior policy. The $D_{KL}$ term in the loss function acts as a regularizer, ensuring the model does not significantly deviate from the distribution where the reward model is most accurate. RL fine-tuning employs reinforcement learning algorithms, like PPO (Schulman et al., 2017), to optimize the above

loss function (Ouyang et al., 2022), introducing significant complexity into the RLHF pipeline. Additionally, the RLHF framework allows for the propagation of any generalization errors from the reward model to the RL fine-tuned model. The DPO framework (Rafailov et al., 2023) addresses these challenges by simplifying the problem into a single-phase supervised learning approach, thus avoiding the pitfalls associated with separate reward modeling and RL fine-tuning phases.

**DPO** circumvents the need for a reward model by directly optimizing the following loss function:

$$\mathcal{L}_{\text{DPO}}(\pi_\theta, \pi_{\text{ref}}, \mathcal{D}) = -\mathbb{E}\left[\log \sigma\left(\beta \log \frac{\pi_\theta(y_w|x)}{\pi_{\text{ref}}(y_w|x)} - \beta \log \frac{\pi_\theta(y_l|x)}{\pi_{\text{ref}}(y_l|x)}\right)\right]. \tag{5}$$

It was demonstrated in (Rafailov et al., 2023) that $\mathcal{L}_{\text{DPO}}$ has the same minimizer as $\mathcal{L}_{\text{RLHF}}$, under the conditions of the BT model, an asymptotically large dataset, and a sufficiently large model capacity (i.e., a *tabular model* that encodes the probability of $\pi_\theta(y|x)$ for all $x \in \mathcal{X}$ and $y \in \mathcal{Y}$ into a vector). The DPO framework was further extended in (Azar et al., 2023), aiming to directly maximize the *win-rate* of $\pi_\theta$ against $\pi_{\text{ref}}$.

## 3 SPO - BASIC

Following (1), we consider a loss function of the form:

$$\mathcal{L}_{\text{SPO}}(\pi_\theta, \pi_{\text{ref}}, \mathcal{D}) = \mathcal{L}_{\text{pref}}(\pi_\theta, \mathcal{D}) + \text{Reg}(\pi_\theta, \pi_{\text{ref}}), \tag{6}$$

where $\mathcal{L}_{\text{pref}}$ and $\text{Reg}$ stand for *preference loss* and *regularizer*, respectively. We proceed to further detail these components.

The regularization term, $\text{Reg}(\pi_\theta, \pi_{\text{ref}})$, aims to ensure that $\pi_\theta$ avoids producing outputs that are highly improbable under $\pi_{\text{ref}}$. A common and effective choice is the KL divergence, $\mathcal{D}_{\text{KL}}(\pi_\theta \parallel \pi_{\text{ref}})$, although other regularization options are viable (Zhao et al., 2022). Importantly, $\text{Reg}(\pi_\theta, \pi_{\text{ref}})$ does not incorporate the preference dataset $\mathcal{D}$ as an input. This is because within $\mathcal{D}$, the model aims to fit to the target preferences, making additional regularization within $\mathcal{D}$ unnecessary. In fact, the regularization term primarily aims to regularize $\pi_\theta$ outside $\mathcal{D}$. This approach diverges from the DPO and several other existing loss functions (detailed in Section 8), which only consider the divergence of $\pi_\theta$ from $\pi_{\text{ref}}$ within the preference dataset.

We now turn our attention to the preference loss. Given a query $x$, let $\pi_\theta(y|x)$ denote the probability that model $\pi_\theta$ generates output $y$. When presented with a query $x$ and two responses, $y_1$ and $y_2$, we define the probability that $\pi_\theta$ prefers $y_1$ over $y_2$ as

$$\mathcal{P}_{\pi_\theta}(y_1 \succ y_2 \mid x) \stackrel{\text{def}}{=} P\big(\text{output of } \pi_\theta(\cdot|x) \text{ is } y_1 \mid \text{output of } \pi_\theta(\cdot|x) \text{ is in } \{y_1, y_2\}\big)$$

$$= \frac{\pi_\theta(y_1|x)}{\pi_\theta(y_1|x) + \pi_\theta(y_2|x)}, \tag{7}$$

where the last equality follows from the definition of conditional probability. We can then employ log-likelihood loss to measure the alignment of preference-probabilities' with the preference-dataset labels,

$$-\mathbb{E}_{(x;y_w,y_l)\sim\mathcal{D}}\big[\log \mathcal{P}_{\pi_\theta}(y_w \succ y_l \mid x)\big]. \tag{8}$$

We consider a preference loss $\mathcal{L}^\alpha_{\text{pref}}(\pi_\theta, \mathcal{D})$ that extends the above cross entropy loss by employing arbitrary exponents for $\pi_\theta$. Specifically, we let for any $\alpha > 0$,

$$\mathcal{L}^\alpha_{\text{pref}}(\pi_\theta, \mathcal{D}) \stackrel{\text{def}}{=} -\frac{1}{\alpha}\mathbb{E}_{(x;y_w,y_l)\sim\mathcal{D}}\left[\log \frac{\pi_\theta(y_w \mid x)^\alpha}{\pi_\theta(y_w \mid x)^\alpha + \pi_\theta(y_l \mid x)^\alpha}\right], \tag{9}$$

and for $\alpha = 0$,

$$\mathcal{L}^0_{\text{pref}}(\pi_\theta, \mathcal{D}) \stackrel{\text{def}}{=} -\frac{1}{2}\mathbb{E}_{(x;y_w,y_l)\sim\mathcal{D}}\left[\log \frac{\pi_\theta(y_w \mid x)}{\pi_\theta(y_l \mid x)}\right]. \tag{10}$$

The $\mathcal{L}^\alpha_{\text{pref}}$ takes the specific form (10) because the gradient of (9) approaches the gradient of (10) when $\alpha \to 0$, as can be easily verified from the following closed form expression for any $\alpha \geq 0$,

$$-\nabla_\theta \mathcal{L}^\alpha_{\text{pref}}(\pi_\theta, \mathcal{D}) = \mathbb{E}_{(x;y_w,y_l)\sim\mathcal{D}}\left[\frac{\pi_\theta(y_l|x)^\alpha}{\pi_\theta(y_w|x)^\alpha + \pi_\theta(y_l|x)^\alpha}\big(\nabla_\theta \log \pi_\theta(y_w|x) - \nabla_\theta \log \pi_\theta(y_l|x)\big)\right].$$

Here, $\pi_\theta(y_l|x)^\alpha / \left(\pi_\theta(y_w|x)^\alpha + \pi_\theta(y_l|x)^\alpha\right)$ serves as a measure of the model's error in preferring $y_w$ over $y_l$. Consequently, the magnitude of this preference error proportionally scales the adjustment $\nabla_\theta \log \pi_\theta(y_w|x) - \nabla_\theta \log \pi_\theta(y_l|x)$, leading to larger updates when the error is large.

The loss function $\mathcal{L}_{\text{pref}}^\alpha(\pi_\theta, \mathcal{D})$ contains the cross-entropy loss in (8) as a special case when $\alpha = 1$. The $\alpha$ parameter allows for tailoring the model to exhibit different entropies; models minimized under $\mathcal{L}_{\text{pref}}^\alpha$ will display higher entropy for larger $\alpha$ values, gradually moving towards a deterministic model akin to DPO as $\alpha$ approaches zero; as established in the next theorem.

Although the SPO framework does not rely on existence of underlying reward functions, and in particular the BT assumption, it is insightful to study the preference loss $\mathcal{L}_{\text{pref}}^\alpha$ under the conditions where the BT model assumption is valid. Intuitively, for a *BT expert model*, defined as $\pi(y|x) = \exp(r(y|x))/Z(x)$ with $Z(x)$ being the partition function, the preference probability in (7) would be identical to the BT preference formula (2). In the next theorem, we further study the landscape of $\mathcal{L}_{\text{pref}}^\alpha$ under the BT model assumption. To eliminate local minima and saddle points that arise from nonlinear model spaces such as neural networks, in the theorems we consider a *tabular model* that encodes the probability of $\pi_\theta(y|x)$ for all $x \in \mathcal{X}$ and $y \in \mathcal{Y}$ into a large vector.

**Theorem 1.** *Suppose that the BT model holds with rewards $r(\cdot|x)$, and fix any probability distribution $\mathcal{D}$ over $\mathcal{X} \times \mathcal{Y} \times \mathcal{Y}$ that has full support[1] and is consistent with the BT assumption.[2] Then, for any $\alpha \geq 0$, in the tabular model, $\mathcal{L}_{\text{pref}}^\alpha$ has a unique minimizer $\text{Softmax}(r(\cdot|x)/\alpha)$ (reducing to $\arg\max r(\cdot|x)$ for $\alpha = 0$). Furthermore, this minimizer is globally absorbing, and the landscape of $\mathcal{L}_{\text{pref}}^\alpha$ contains no other first-order stationary point (i.e., no other local minima, local maxima, or saddle points).*

The proof is provided in Appendix A. According to Theorem 1, minimizer of $\mathcal{L}_{\text{pref}}^\alpha$ is the softmax of BT rewards divided by $\alpha$, where $\alpha$ controls the entropy of the final model. Specifically, in the the asymptotically large dataset regime, when $\alpha = 1$, the preference loss reaches its minimum at the hypothetical *BT expert model* that generates the preference dataset's labels, defined as $\text{Softmax}(r(\cdot|x))$.

# 4 THE GENERAL SPO ALGORITHM

We further expand the preference loss of SPO by considering a weighting over different samples, where the weights can depend on $\pi_\theta$. This weighting only affects (improves) the optimization process without changing the fixed point, as we show in this section.

We call a function $\mu : \mathcal{Y} \times \mathcal{Y} \times \mathcal{X} \to \mathbb{R}^+$ *symmetric positive* if $\mu(y_1, y_2 \mid x) = \mu(y_2, y_1 \mid x) > 0$, for all $x \in \mathcal{X}$ and all $y_1, y_2 \in \mathcal{Y}$. Given a symmetric positive function $\mu$ and an $\alpha \geq 0$, we define *weighted* preference loss as

$$\mathcal{L}_{\text{pref}}^{\alpha,\mu}(\pi_\theta, \mathcal{D}) \stackrel{\text{def}}{=} -\frac{1}{\alpha} \mathbb{E}_{(x;y_w,y_l)\sim\mathcal{D}} \left[ \mu(y_w, y_l \mid x) \log \frac{\pi_\theta(y_w \mid x)^\alpha}{\pi_\theta(y_w \mid x)^\alpha + \pi_\theta(y_l \mid x)^\alpha} \right] \quad (11)$$

if $\alpha > 0$, and for $\alpha = 0$ we let

$$\mathcal{L}_{\text{pref}}^{0,\mu}(\pi_\theta, \mathcal{D}) \stackrel{\text{def}}{=} -\frac{1}{2} \mathbb{E}_{(x;y_w,y_l)\sim\mathcal{D}} \left[ \mu(y_w, y_l \mid x) \log \frac{\pi_\theta(y_w \mid x)}{\pi_\theta(y_l \mid x)} \right]. \quad (12)$$

The weight-function $\mu$ controls the impact of individual samples within the loss calculation. The utility of $\mu$ emerges from the observation that not all sample pairs in the preference dataset hold equivalent significance. For instance, diminishing the weights of dataset samples $(x; y_w, y_l)$ where both responses $y_w$ and $y_l$ are of low quality (e.g., low probability) can be particularly advantageous. This can be achieved for example by setting

$$\mu(y_1, y_2 \mid x) = 2\,\sigma\Big( \big(\pi_\theta(y_1|x) + \pi_\theta(y_2|x)\big)^\gamma - \hat{\mathbb{E}}_{(y_1',y_2'|x')\sim\mathcal{D}} \left[ \big(\pi_\theta(y_1'|x') + \pi_\theta(y_2'|x')\big)^\gamma \right] \Big), \quad (13)$$

---

[1] Full support in this context means that the probability distribution assigns a non-zero sampling probability to all $(x; y_w, y_l) \in \mathcal{X} \times \mathcal{Y} \times \mathcal{Y}$.

[2] Consistency with the BT holds if the relative probability of outcomes is determined by a logistic function of the reward differences. More specifically, $\mathcal{D}(x; y_1, y_2)/\mathcal{D}(x; y_2, y_1) = p^{\text{BT}}(y_1 \succ y_2|x)/p^{\text{BT}}(y_2 \succ y_1|x) = \exp\big(r(y_1 \mid x) - r(y_2 \mid x)\big)$, for all $(x; y_1, y_2) \in \mathcal{X} \times \mathcal{Y} \times \mathcal{Y}$, where $p^{\text{BT}}$ is defined in (2) and $r(\cdot|\cdot)$ is the reward function in the BT model.

---

**Algorithm 1** SPO

---
   **for** $t = 0, 1, 2, \ldots$ **do**
      **if** $t$ is a multiple of $T$ **:**       # once every $T$ iterations
           Generate a batch $\mathcal{B}$ of online samples $y \sim \pi_\theta(\cdot|x)$, for a set of recently observed $x \sim \mathcal{D}$.
       Compute $\mathcal{L}_{\text{pref}}^{\alpha,\mu}(\pi_\theta, \mathcal{D})$ from (11), using the $\mu$ given in (13).
       Compute token-wise regularizer $\widehat{\mathcal{D}_{\text{KL}}}(\pi_\theta \parallel \pi_{\text{ref}})$ from (15), using the online samples batch $\mathcal{B}$.
       Form the SPO loss function $\mathcal{L}_{\text{SPO}}(\pi_\theta, \pi_{\text{ref}}, \mathcal{D}) = \mathcal{L}_{\text{pref}}^{\alpha,\mu}(\pi_\theta, \mathcal{D}) + \widehat{\mathcal{D}_{\text{KL}}}(\pi_\theta \parallel \pi_{\text{ref}})$.
       Update the network using an optimizer of interest for the loss function $\mathcal{L}_{\text{SPO}}(\pi_\theta, \pi_{\text{ref}}, \mathcal{D})$.
   **end for**

---

where $\sigma$ is the sigmoid function, $\gamma \geq 0$ is a hyperparameter (e.g., 0.01) to dampen the influence of exponentially shrinking probabilities, and $\hat{\mathbb{E}}$ aims to ensure that the significance of a pair is measured relative to other pairs and can be obtained by averaging over the current batch. The $\mu$ function boils down to uniform weights if $\gamma = 0$ or if all pairs in the batch have similar sum-probabilities.

While $\mu$ may depend on $\pi_\theta$, it is important to note that gradient propagation through $\mu$ is not permitted. Specifically, the gradient $\nabla_\theta \mathcal{L}_{\text{pref}}^{\alpha,\mu}(\pi_\theta, \mathcal{D})$ is given by

$$
-\mathbb{E}_{(x;y_w,y_l)\sim\mathcal{D}} \left[ \mu(y_w, y_l|x) \frac{\pi_\theta(y_l|x)^\alpha}{\pi_\theta(y_w|x)^\alpha + \pi_\theta(y_l|x)^\alpha} \left( \nabla_\theta \log \pi_\theta(y_w|x) - \nabla_\theta \log \pi_\theta(y_l|x) \right) \right].
\tag{14}
$$

Interestingly, the weight function, $\mu$, mainly influences the optimization process, not the ultimate fixed point, in the tabular setting and under asymptotically large preference dataset, as we show in the next theorem. The proof is given in Appendix A.

**Theorem 2.** *Suppose that the conditions of Theorem 1 hold. Then for any $\alpha \geq 0$ and any symmetric positive function $\mu$, the softmax of the BT rewards divided by $\alpha$, $\text{Softmax}(r(\cdot|x)/\alpha)$ (reducing to $\arg\max r(\cdot|x)$ for $\alpha = 0$), is the unique globally absorbing fixed point of the differential equation $\dot{\pi} = \prod \left( -\nabla_\theta \mathcal{L}_{\text{pref}}^{\alpha,\mu}(\pi_\theta, \mathcal{D}) \right)$, where $\prod(\cdot)$ stands for projection onto the probability simplex, and the gradient is given in* (14).

We now proceed to discuss the computation of $\mathcal{D}_{\text{KL}}$ regularizer in (6). In order to estimate $\mathcal{D}_{\text{KL}}(\pi_\theta \| \pi_{\text{ref}}) = \mathbb{E}_x \mathbb{E}_{y\sim\pi_\theta(\cdot|x)} \left[ \log \left( \pi_\theta(y|x)/\pi_{\text{ref}}(y|x) \right) \right]$, we generate online samples from the current model $\pi_\theta$. This is however costly for sequential models, where sequence generation necessitates sequential calls to the model. To mitigate this problem, we generate a batch of samples from $\pi_\theta$ intermittently, for example once every $T$ steps, and keep using samples from this batch for approximating $\mathcal{D}_{\text{KL}}$, until the next batch of samples is generated.

Given a batch of samples $(x, y)$ with $y \sim \pi_\theta(\cdot|x)$, in order to we obtain a reduced variance approximation of $\mathcal{D}_{\text{KL}}$, we employ the following token-wise $\mathcal{D}_{\text{KL}}$ formula:

$$
\begin{aligned}
\widehat{\mathcal{D}_{\text{KL}}}(\pi_\theta \parallel \pi_{\text{ref}}) &\stackrel{\text{def}}{=} \mathbb{E}_{(x;y)\in\text{batch}} \left[ \sum_{\tau=1}^{|y|} \mathcal{D}_{\text{KL}}\left( \pi_\theta(Y_\tau \mid x, y_{:\tau}) \parallel \pi_{\text{ref}}(Y_\tau \mid x, y_{1:\tau}) \right) \right] \\
&= \mathbb{E}_{(x;y)\in\text{batch}} \left[ \sum_{\tau=1}^{|y|} \sum_{s\in\mathcal{S}} \pi_\theta(Y_\tau = s \mid x, y_{1:\tau}) \log \frac{\pi_\theta(Y_\tau = s \mid x, y_{1:\tau})}{\pi_{\text{ref}}(Y_\tau = s \mid x, y_{1:\tau})} \right],
\end{aligned}
\tag{15}
$$

where $\mathcal{S}$ is the set of all possible tokens. Note that $\pi(Y_\tau = s \mid x, y_{1:\tau})$ is readily available from the softmax of the logits, in the network's output. Therefore, the sum in (15) can be computed with negligible computational overhead (excluding the initial forward path). Note that $\widehat{\mathcal{D}_{\text{KL}}}$ is a biased estimate of the sequence-$\mathcal{D}_{\text{KL}}$. However, we empirically found that the benefit of reduced variance brought by the token-wise approximation out-weights the potential negative impact of the resulting bias. Moreover, similar to sequence-$\mathcal{D}_{\text{KL}}$, the token-wise $\mathcal{D}_{\text{KL}}$ is a proximity measure for $\pi_\theta$ and $\pi_{\text{ref}}$, and is therefore a conceptually sound choice of regularizer. Algorithm 1 summarized the SPO algorithm.

## 5 SPO FOR OTHER DATA-TYPES: BEST-OF-$n$ PREFERENCE AND RANKED PREFERENCE

In this section, we generalize the SPO algorithm for other types of preference data: best-of-$n$ preference data and ranked-data. We extend the definition of a symmetric function to $n$-responses by calling a function $\mu : \mathcal{Y}^n \times \mathcal{X} \to \mathbb{R}^+$ *symmetric positive* if $\mu(y_{\tau(1)}, \ldots, y_{\tau(n)} \mid x) = \mu(y_1, \ldots, y_n \mid x) > 0$, for all $x \in \mathcal{X}$ all $y_1, \ldots, y_n \in \mathcal{Y}$, and all permutations $\tau$ of $(1, \ldots, n)$.

**Best-of-$n$ preference data:** Given an $n \geq 2$, a sample $(x; y_1, \ldots, y_n; i^*)$ of a best-of-$n$ preference dataset consists of a query $x$ along with $n$ responses $y_1, \ldots, y_n$, one of which (i.e., $y_{i^*}$) is labeled by the expert as the best response. Given a symmetric positive function $\mu$ and an $\alpha > 0$, we propose the following preference loss for a best-of-$n$ preference dataset $\mathcal{D}$:

$$\mathcal{L}^{\alpha,\mu}_{\text{pref-}n}(\pi_\theta, \mathcal{D}) \stackrel{\text{def}}{=} -\frac{1}{\alpha} \mathbb{E}_{(x;y_1,\ldots,y_n;i^*)\sim\mathcal{D}} \left[ \mu(y_1, \ldots, y_n \mid x) \, \log \frac{\pi_\theta(y_{i^*} \mid x)^\alpha}{\sum_{i=1}^n \pi_\theta(y_i \mid x)^\alpha} \right]. \tag{16}$$

As before, we stop the gradient from propagating through $\mu$, even though $\mu$ may depend on $\pi_\theta$. Similar to the case of pairwise preferences, we show in the following theorem that the loss in (16) is minimized at the softmax of rewards, if we assume existence of an underlying reward function. In particular, given a reward function $r(\cdot|x) : \mathcal{X} \to \mathcal{Y}$ and a distribution $\mathcal{D}$ over $\mathcal{X} \times \mathcal{Y}^n \times \{1, \ldots, n\}$, we say that $\mathcal{D}$ is *consistent with $n$-ary BT model* if for any $(x; y_1, \ldots, y_n) \in \mathcal{X} \times \mathcal{Y}^n$ and any $i, j \in \{1, \ldots, n\}$, $\mathcal{D}(x; y_1, \ldots, y_n; i)/\mathcal{D}(x; y_1, \ldots, y_n; j) = \exp\left(r(y_i \mid x) - r(y_j \mid x)\right)$. Note that this definition boils down to the definition of consistency with BT model for $n = 2$ in Section 3. Proof of the following theorem is given in Appendix B.

**Theorem 3.** *Consider a reward function $r(\cdot \mid x)$ and a probability distribution $\mathcal{D}$ with full support over $\mathcal{X} \times \mathcal{Y}^n \times \{1, \ldots, n\}$ that is consistent with the $n$-ary BT model. Then, for any $\alpha > 0$ and any symmetric positive function $\mu$, in the tabular model, $\text{Softmax}(r(\cdot|x)/\alpha)$ is the unique globally absorbing fixed point of the differential equation $\dot{\boldsymbol{\pi}} = \prod \left( -\nabla_\theta \mathcal{L}^{\alpha,\mu}_{\text{pref-}n}(\pi_\theta, \mathcal{D}) \right)$, where $\prod(\cdot)$ stands for projection onto the probability simplex.*

**Ranked Preference Data:** A ranked preference dataset consists of samples of the form $(x; y_1, \ldots, y_n; \tau)$, where $x$ is a query, $y_1, \ldots, y_n$ are $n$ responses, and $\tau$ is a permutation representing the relative preference $y_{\tau(1)} \succ \cdots \succ y_{\tau(n)}$ of the expert over these responses. Given an $\alpha > 0$ and a sequence of symmetric positive function $\mu_k : \mathcal{X} \times \mathcal{Y}^k \to \mathbb{R}$ for $k = 2, \ldots, n$, we propose the following preference loss for a ranked preference dataset $\mathcal{D}$:

$$\mathcal{L}^{\alpha,[\mu]}_{\text{rank}}(\pi_\theta, \mathcal{D}) \stackrel{\text{def}}{=} -\frac{1}{\alpha} \mathbb{E}_{(x;y_1,\ldots,y_n;\tau)\sim\mathcal{D}} \left[ \sum_{k=1}^{n-1} \mu_k(y_{\tau(k)}, \ldots, y_{\tau(n)}|x) \, \log \frac{\pi_\theta(y_{\tau(k)} \mid x)^\alpha}{\sum_{j=k}^n \pi_\theta(y_{\tau(j)} \mid x)^\alpha} \right]. \tag{17}$$

We can control the importance weight of responses in different ranks through appropriate adjustment of weight functions $\mu^1, \ldots, \mu^{n-1}$. For example, by setting $\mu_k = 0$ for $k = 2, \ldots, n-1$, $\mathcal{L}^{\alpha,[\mu]}_{\text{rank}}$ boils down to $\mathcal{L}^{\alpha,\mu^1}_{\text{pref-}n}$. Here again, the gradient is not allowed to propagate through $\mu^1, \ldots, \mu^{n-1}$, even though these functions may depend on $\pi_\theta$. The following theorem shows that, assuming existence of underlying rewards under the PL model (3), the softmax of these rewards is the unique minimizer of $\mathcal{L}^{\alpha,\mu}_{\text{rank}}$. The proof relies on Theorem 3, and is given in Appendix C.

**Theorem 4.** *Suppose that the PL model holds with rewards $r(\cdot|x)$, and a probability distribution $\mathcal{D}$ with full support over $\mathcal{X} \times \mathcal{Y}^n \times \{\text{Identity permutation}\}$ that is consistent with the PL model.[3] Then, for any $\alpha > 0$ and any sequence $[\mu] = \mu^1, \ldots, \mu^{n-1}$ of symmetric positive functions, in the tabular model, $\text{Softmax}(r(\cdot|x)/\alpha)$ is the unique globally absorbing fixed point of the differential equation $\dot{\boldsymbol{\pi}} = \prod \left( -\nabla_\theta \mathcal{L}^{\alpha,[\mu]}_{\text{rank}}(\pi_\theta, \mathcal{D}) \right)$, where $\prod(\cdot)$ stands for projection onto the probability simplex.*

---

[3]Consistency with the PL model holds if $\mathcal{D}(x; y_1, \ldots, y_n; \tau)/\mathcal{D}(x; y_1, \ldots, y_n; \tau') = p^{\text{PL}}(y_{\tau(1)} \succ \cdots \succ y_{\tau(n)}|x)/p^{\text{PL}}(y_{\tau'(1)} \succ \cdots \succ y_{\tau'(n)}|x)$, for all $(x; y_1, \ldots, y_n) \in \mathcal{X} \times \mathcal{Y}^n$ and all permutations $\tau$ and $\tau'$, where $p^{\text{PL}}$ is defined in (3).

## 6 COMPARATIVE ANALYSIS: SPO VERSUS DPO

This section contrasts the SPO method with the DPO algorithm conceptually. A detailed empirical comparison follows in Section 7.

A key difference between SPO and DPO lies in how regularization ($\mathcal{D}_{\mathrm{KL}}$) is applied. In RLHF and SPO, $\mathcal{D}_{\mathrm{KL}}$ is used to prevent $\pi_\theta$ from straying too far from $\pi_{\mathrm{ref}}$ in unexplored regions, reducing the risk of distribution shifts. In contrast, the DPO loss function (5) applies regularization only to preference dataset samples, which is suboptimal because 1) it fails to prevent distribution shifts in unexplored regions, and 2) regularizing within the dataset can hinder alignment with the preferences. In contrast, SPO incorporates a global regularizer to prevent undesired out-of-dataset distribution shifts. In the context of a question-answering task, the term "out-of-dataset region" refers to query-response pairs where the response is not part of the preference dataset. The query in these pairs may or may not be included in the dataset, but it should, in either case, be relevant to the test domain.

Moreover, SPO has an advantage over DPO and RLHF in avoiding determinism. In cases where the preference dataset is comparable to pre-training data size, regularization ($\mathcal{D}_{\mathrm{KL}}$) becomes unnecessary (and we can set $\beta \simeq 0$), in which case RLHF and DPO loss functions tend to produce deterministic models; that is they tend to return a single high-quality response per query. This reduced diversity makes DPO prone to mode collapse (Azar et al., 2023). SPO, however, controls entropy via the $\alpha$ parameter in (9), even without the use of regularizer (see Theorem 1), preserving response diversity. This makes SPO more adaptable for continual learning and future alignments.

It is noteworthy that unlike RLHF and DPO, the SPO framework does not assume the existence of an underlying reward model or rely on assumptions like the BT model. Instead, SPO's preference loss directly aligns $\pi_\theta$ with the preferences in the dataset, making it potentially more adaptable to broader alignment contexts. Additionally, SPO is not limited to using $\mathcal{D}_{\mathrm{KL}}$ for regularization, unlike DPO and IPO, which depend on $\mathcal{D}_{\mathrm{KL}}$ for derivations of their loss functions.

We also note that the DPO loss cannot be separated into components like (6), where the preference loss is independent of $\pi_{\mathrm{ref}}$ and paired with a regularizer like $\mathcal{D}_{\mathrm{KL}}$. As a short proof, consider a case where $\pi_\theta(y_w|x) = \pi_\theta(y_l|x)$ for a sample $(x; y_w, y_l) \in \mathcal{D}$. Here, the alignment loss in (6) remains symmetrical with respect to $\pi_{\mathrm{ref}}(y_w|x)$ and $\pi_{\mathrm{ref}}(y_l|x)$; swapping their values wouldn't affect either the preference loss or $\mathcal{D}_{\mathrm{KL}}$. This symmetry doesn't hold in the DPO framework, as seen in the DPO loss in (5). Therefore, the DPO loss cannot be represented in the separable form of (6).

## 7 EXPERIMENTS

This section presents empirical evaluations of SPO. All codes and datasets are available online at (Anonymous, 2024).

### 7.1 ALIGNMENT TO PAIRWISE PREFERENCE DATA

**Experiment setting:** To evaluate the performance of SPO, we trained a Llama2-7B model (Touvron et al., 2023) on a pairwise preference dataset for question-answering available in AlpacaFarm (Dubois et al., 2023), and computed the win-rates against the Llama2-7B SFT model on AlpacaEval 2 (Li et al., 2023), using GPT4-Turbo API. More specifically, we used the following pipeline. We downloaded the pretrained Llama2-7B model and performed supervised fine-tuning on the AlpacaFarm SFT dataset available at Dubois et al. (2023), to obtain the SFT model. Initializing the weights on the SFT model, we then performed alignment to the preference dataset from AlpacaFarm that contains pairs of question-answering samples and preference labels provided by GPT-4. We compared the performance of SPO with several reward-model-free alignment algorithms, namely DPO (Rafailov et al., 2023), IPO (Azar et al., 2023), KTO (Ethayarajh et al., 2024), CPO (Xu et al., 2024), R-DPO (Park et al., 2024), and SimPO (Meng et al., 2024). For SPO, the experiment includes both the basic and weighted versions of SPO, with the weight function $\mu$ given in (13). We trained each algorithm for a few epochs, used GPT4-Turbo to compute its win-rate against the SFT model at the end of each epoch, and reported the maximum win-rate for each algorithm. Additional details about setting of this experiment is provided in Appendix D.1.

**Results:** Table 1 presents the win-rates and length-controlled (LC) win-rates (Dubois et al., 2023) of different algorithms against the SFT model. The SPO algorithm outperforms the other tested algorithms in both conventional win-rate and LC win-rate. Furthermore, the weighted version of SPO performs better than the basic version, demonstrating the advantage of incorporating the weight function. Standard deviation of the reported win-rates is 1.5%. Moreover, LC win-rates in Table 1 are evaluated on models with hyperparameters optimized for win-rates. Notably, SPO shows better generalization to LC win-rates compared to other baselines, some of which show less than 50% LC winrate against the SFT model.

Table 1: Alignment of the Llama2-7B model on AlpacaFarm dataset.

| Alignment method | Win-rate(%) | LC Win-rate(%) |
|---|---|---|
| SFT | 50.00 | 50.00 |
| R-DPO | 52.50 | 41.49 |
| CPO | 54.10 | 39.38 |
| SimPO | 58.48 | 49.73 |
| KTO | 58.50 | 51.94 |
| IPO | 58.59 | 49.60 |
| DPO | 59.16 | 51.26 |
| SPO-basic (unweighted) | 60.83 | 53.17 |
| **SPO** | **61.63** | **56.25** |

## 7.2 ALIGNMENT TO RANKING AND BEST-OF-$n$ PREFERENCE DATA

**Experiment setting:** *Dataset:* Alignment research has largely focused on pairwise preferences, and publicly available datasets for other preference types are rather scarce. To address this, we generated an $n$-ary ranked preference dataset ($n = 4$) using GPT4o API as the labeler. Building on TinyStories (Eldan & Li, 2023) –a synthetic collection of short stories aimed at children aged 3 to 4– we created a preference dataset to align the stories to an older audience. The dataset contains around 5,000 samples, each with four stories generated by the TinyStories pre-trained 110M model, ranked by GPT4o-2024-08-06 based on coherence and engagement for older students. The best-of-$n$ dataset was then derived by removing the rank labels for the 2nd, 3rd, and 4th responses. Further details are in Appendix E, and the datasets are available online at (Anonymous, 2024).

*Training:* Using the implementation from (Karpathy, 2024) and the supervised fine-tuned model from (Karpathy, 2023), we aligned a 110M parameter model with ranking and best-of-$n$ versions of SPO. We compared this with three baselines: the ranking version of DPO (AppendixA3 of (Rafailov et al., 2023)), S-DPO (Chen et al., 2024), and Best-response-SFT. Best-response-SFT is derived by all performing supervised fine-tuning on all top-rank responses. See Appendix D.2 for further details.

**Results:** Tables 2 and 3 present peak win rates against the Best-response-SFT model for aligning the 110M TinyStories SFT model to ranking and best-of-$n$ datasets using different algorithms. The SPO algorithm outperforms all baselines for both types of alignment. Standard deviation of the reported win-rates is 1.4%.

Table 2: Alignment of TinyStories SFT model to the ranking dataset.

| Alignment method | Win-rate (%) |
|---|---|
| Best-response-SFT | 50.0 |
| DPO (ranking version) | 67.0 |
| S-DPO (ranking) | 55.1 |
| SPO (ranking) | **68.5** |

Table 3: Alignment of TinyStories SFT model to the best-of-$n$ dataset.

| Alignment method | Win-rate (%) |
|---|---|
| Best-response-SFT | 50.0 |
| S-DPO (best-of-$n$) | 66.0 |
| SPO-basic (best-of-$n$) | 67.8 |
| SPO (best-of-$n$) | **70.5** |

## 7.3 ABLATION STUDY

**Global regularization vs in-dataset regularization:** To evaluate the significance of global regularization, we trained a Llama2-7B SFT model the SPO-basic algorithm while replacing the global $\mathcal{D}_{\text{KL}}$

regularizer of (15) with in-dataset regularizers. In particular, we tested three different in-dataset regularizers, including the popular $-\log \pi(y_w|x)$ regularizer that is used in several alignment algorithm like CPO, SLiC-HF (Zhao et al., 2023), and RRHF (Yuan et al., 2024). Other experiment settings are similar to the setting discussed in Section 7.1. Table 4 provides the list of these in-dataset regularizers, as well as the peak win-rates of SPO-basic using these regularizers against the Llama2-7B SFT model. As it can be seen from the table, the global $\mathcal{D}_{\mathrm{KL}}$ regularizer of (15) achieved better performance compared to in-dataset regularizers.

**Weight function** $\mu$**:** As observed in the experiment of Section 7.1 on the alignment to AlpacaFarm preferences dataset, the weighted version of SPO achieved win-rate $61.63\%$ compared to the win-rate of $60.83\%$ for SPO-basic. In the experiment of Section 7.2 on the alignment to best-of-$n$ preferences dataset, the weighted version of SPO improved the win-rate to $70.5\%$, up from $67.8\%$ for SPO-basic. In the ranking experiment, use of non-uniform $\mu$ did not improve the win-rate. In all of these experiments, the $\mu$ function in the weighted SPO algorithm is of the form (13) with parameter $\gamma = 0.01$. We performed no sweeping on the parameter $\gamma = 0.01$.

Table 4: Ablation of SPO regularizer (model: Llama2-7B, dataset: AlpacaFarm)

| Regularizer type | Regularizer formula | Win-rate vs. SFT |
|---|---|---|
| in-dataset (log probability) | $-\mathbb{E}_{(x;y_w,y_l)\in\mathcal{D}}\big[\log \pi_\theta(y_w|x)\big]$ | 54.1 |
| in-dataset (tokenwise) | $\mathbb{E}_{(x;y)\in D}\left[\sum_{\tau=0}^{|y|-1}\mathbb{E}_{Y\sim\pi_\theta(\cdot|x,y_{1:\tau})}\log\frac{\pi_\theta(Y|x,y_{1:\tau})}{\pi_{\mathrm{ref}}(Y|x,y_{1:\tau})}\right]$ | 59.7 |
| in-dataset (importance sampling) | $\mathbb{E}_{(x;y)\in D}\left[\frac{\pi_\theta(y|x)}{\pi_{\mathrm{ref}}(y|x)}\log\frac{\pi_\theta(y|x)}{\pi_{\mathrm{ref}}(y|x)}\right]$ | 59.9 |
| global | Eq. (15) | **60.8** |

# 8 Related Works

RLHF aims to align AI systems with human preferences, relying on human judgments rather than manual rewards or demonstrations. This method has been successfully applied in fine-tuning large language models (LLMs) (Achiam et al., 2023; Touvron et al., 2023; Ouyang et al., 2022), but faces challenges including data quality issues, reward misgeneralization, and policy optimization complexities. Research to enhance RLHF includes methods such as rejection sampling for response generation (Dong et al., 2023; Touvron et al., 2023), where the highest-reward response from a fixed number is selected for fine-tuning. Zhang et al. (2023) simplified instruction alignment with language models into a goal-oriented reinforcement learning task, utilizing a two-phase approach of high-temperature online sampling and supervised learning with relabeled data during offline training. A two-loop learning algorithm, Grow and Improve, has also been proposed for iterative model alignment and training on a fixed dataset (Gulcehre et al., 2023). The Grow loop leverages the existing model to create and sample a dataset while the Improve loop iteratively trains the model on a fixed dataset.

Given the challenges of RLHF, reward-model-free alignment methods emerged fairly recently and have gained a lot of popularity. Reward-model-free approach to alignment was popularized specifically after introduction of DPO in (Rafailov et al., 2023), which is breifly outlined in Section 2. Recently, several works have been proposed methods to improve DPO. Building on DPO, Ji et al. (2024) proposed Efficient Exact Optimization (EXO) method, which aligns language models by minimizing reverse KL divergence, offering a more stable and efficient alternative to RL-based methods.Azar et al. (2023) proposed an objective called $\Psi$PO for learning from human preferences that is expressed in terms of pairwise preferences, with no need for assumption of the BT model. The authors focused on a specific instance, IPO, of $\Psi$PO by setting $\Psi$ as the identity, aiming to mitigate the overfitting and tendency-towards-deterministic-policies issues observed in DPO. Munos et al. (2023) formulated the alignment problem as finding the Nash equilibrium (NE) of a maximin game with two policies $\pi$ and $\pi'$ as two players where each policy receives pay off of probability of winning over the other policy. The authors showed that the NE point can be approximated by running a mirror-descent-like algorithm. Rosset et al. (2024) and Swamy et al. (2024) proposed other approaches to approximate the NE point based on no-regret algorithms (Freund & Schapire, 1997). Chowdhury et al. (2024) proposed a loss function which is an unbiased estimate of the original DPO

loss, and aims to alleviate sensitivity to flipped labels due to labeling noise. Amini et al. (2024) added an offset term within the sigmoid function in the DPO loss. In this manner, the model puts more weight on the winning response. To control the length of the output, R-DPO (Park et al., 2024) modified the DPO loss by adding a regularization term that penalizes lengthy responses. To achieve the same goal, SimPO (Meng et al., 2024) replaced the log-likelihood ratio between the current policy and the baseline model with the average log probability of the sequence under the current policy. In (Rafailov et al., 2024), a token-level formulation of DPO has been proposed which enables a likelihood search over a DPO model by classical search-based algorithms, such as MCTS. Inspired by cringe loss previously proposed for binary feedback, Xu et al. (2023) adapted cringe loss for the pairwise preference context. Recently, in (Ethayarajh et al., 2024), KTO loss has been proposed for alignment from non-paired preference datasets. Sessa et al. (2024) introduced BOND, which replaces a reward model with a win-rate model that predicts the likelihood of a response being better than a randomly sampled reference response. This win-rate model is then used to train the policy towards a best-of-$n$ policy, effectively aligning it with human preferences. In the same vein, Gui et al. (2024) proposed BoNBoN Alignment, which directly integrates best-of-$n$ sampling into the training process,

In practice, the performance of an alignment technique highly depends on the quality of the human preference dataset. Noisy preference pairs could potentially limit the language models from capturing human intention. In (Liu et al., 2023), DPO was used in conjunction with an improved preference dataset via a rejection sampling technique, arguing that DPO suffers from a mismatch between the sampling distribution and the policy corresponding to true expert preferences. (Tunstall et al., 2023) formed a dataset of conservative pairs by collecting AI feedback through an ensemble of chat model completions, followed by GPT-4 scoring. Then, they employed DPO for alignment to this improved dataset. The work in (Yin et al., 2024) leveraged semantic correlations of prompts in the dataset to form more conservative response pairs. For a given prompt $(x; y_w, y_l)$, a prompt $x'$ with a similar semantic to a tuple $(x'; y'_w, y'_l)$ is used to form more conservative pairs.

Zhao et al. (2022) proposed a separable alignment technique called SLiC, where, similar to SPO, the alignment loss is the sum of two terms: a calibration loss that contrasts winner and loser responses, encouraging the model $\pi_\theta$ to assign more probability to the winner, and a regularizer term. SLiC was further developed in (Zhao et al., 2023) for alignment to preference data, where they proposed the SLiC-HF algorithm. SLiC-HF involves a rectified contrastive loss as its calibration loss and a log-likelihood term as the regularization. Unlike SPO, SLiC-HF's regularization is limited to the preference or pre-training datasets, not using online samples from $\pi_\theta$ as in the $\mathcal{D}_{\text{KL}}$ regularizer.

Concurrent to our work, CPO (Xu et al., 2024) motivated the preference loss in (9) as a heuristic approximation of DPO to reduce its complexity by removing $\pi_{\text{ref}}$ from the DPO loss. This results in a contrastive loss similar to SPO-basic (9). In contrast, we derived (9) and its extension in (11) as what should be truly minimized, regardless of complexity considerations. Other major differences include CPO's use of in-dataset regularization versus SPO's global regularization, the incorporation of a weighting mechanism in SPO, the generalization of SPO to other data types, and theoretical guarantees proposed in this work. As we demonstrated in experiments (see Table 4), the choice of regularizer significantly impacts performance.

## 9 LIMITATIONS

This paper introduced SPO, a class of algorithms designed for alignment to the expert's distribution, with controlled softness. The paper also presents theoretical results demonstrating favorable landscape and convergence properties of SPO. Below, we discuss some limitations of this work.

**Limitations of the SPO Algorithm:** The main limitation of the SPO framework is the computational complexity of the regularizer, which requires online sampling from $\pi_\theta$. This limitation was discussed in Section 4, when intermittent batch generation of samples was proposed to mitigate the computational overhead .

**Limitations of the Current Study:** The "softness" of the model's output, controlled by the parameter $\alpha$, allows SPO to excel in applications requiring exploration and diversity, such as LLM reasoning. Since current alignment benchmarks prioritize win rates over diversity, our experiments may underestimate SPO's potential, exploration of which remains a subject for future research.

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

# Appendices

## A  PROOF OF THEOREMS 1 AND 2

In this appendix, we present the proof of Theorems 1 and 2. The high-level proof idea is to show that moving along the projected[4] negative gradient of the preference loss (i.e., the ODE direction) results in an absolute reduction of the Euclidean distance of $\pi_\theta$ from $\mathrm{Softmax}\left(r(\cdot|x)/\alpha\right)$.

Without loss of generality, we prove the theorem for a single fixed $x \in \mathcal{X}$, and remove $x$ from the notations, for the sake of notation simplicity.

Given the rewards $r(\cdot)$ in the Bradley-Terry model, let

$$\pi^*(\cdot) \overset{\text{def}}{=} \mathrm{Softmax}\left(r(\cdot)\right). \tag{18}$$

For any $\alpha \in [0, 1)$, let

$$\pi_\alpha^*(\cdot) \overset{\text{def}}{=} \mathrm{Softmax}\left(r(\cdot)/\alpha\right), \tag{19}$$

and let $\boldsymbol{\pi}^\alpha$ be its vector representation. Therefore, for any $\alpha \in [0, 1]$, and for any $y$,

$$\pi^*(y) = z_\alpha \times \left(\pi_\alpha^*(y)\right)^\alpha, \qquad \text{where } z_\alpha \overset{\text{def}}{=} \frac{\left(\sum_{y'} e^{r(y')/\alpha}\right)^\alpha}{\sum_{y'} e^{r(y')}}. \tag{20}$$

Moreover, it follows from the consistency of distribution $\mathcal{D}$ with the Bradley-Terry model that for any pair $(y_1, y_2)$,

$$\frac{\mathcal{D}(y_1, y_2)}{\mathcal{D}(y_1, y_2) + \mathcal{D}(y_2, y_1)} = \mathcal{P}_\mathcal{D}\left(y_1 \succ y_2\right) = \frac{\exp r(y_1)}{\exp r(y_1) + \exp r(y_2)} = \frac{\pi^*(y_1)}{\pi^*(y_1) + \pi^*(y_2)}. \tag{21}$$

For any $y_1, y_2 \in \mathcal{Y}$ let

$$\tilde{\mu}(y_1, y_2) \overset{\text{def}}{=} \mu(y_1, y_2)\left(\mathcal{D}(y_1, y_2) + \mathcal{D}(y_2, y_1)\right). \tag{22}$$

Note that the symmetry of $\mu$ implies symmetry of $\tilde{\mu}$ with respect to its first and second arguments. Then,

$$\mu(y_1, y_2)\,\mathcal{D}(y_1, y_2) = \tilde{\mu}(y_1, y_2)\,\frac{\mathcal{D}(y_1, y_2)}{\mathcal{D}(y_1, y_2) + \mathcal{D}(y_2, y_1)} = \tilde{\mu}(y_1, y_2)\,\frac{\pi^*(y_1)}{\pi^*(y_1) + \pi^*(y_2)}, \tag{23}$$

where the last equality follows from (21).

Consider a $\boldsymbol{\pi}_\theta$ in the relative interior of the probability simplex and let $\mathbf{v}$ be the negative gradient of the preference loss

$$\mathbf{v} \overset{\text{def}}{=} -\nabla_{\pi_\theta}\,\mathcal{L}_{\mathrm{pref}}^{\alpha,\mu}(\pi_\theta, \mathcal{D}), \tag{24}$$

where $\mathcal{L}_{\mathrm{pref}}^{\alpha,\mu}$ is defined in (14). For any $y \in \mathcal{Y}$, let $v(y)$ be the entry of $\mathbf{v}$ that corresponds to $y$. Then,

$$
\begin{aligned}
v(y) &= \sum_{y' \in \mathcal{Y}} \mathcal{D}(y, y')\mu(y, y')\,\frac{\pi_\theta(y')^\alpha}{\pi_\theta(y)^\alpha + \pi_\theta(y')^\alpha}\left(\frac{d}{d\,\pi_\theta(y)}\log \pi_\theta(y) - \frac{d}{d\,\pi_\theta(y)}\log \pi_\theta(y')\right) \\
&\quad + \sum_{y' \in \mathcal{Y}} \mathcal{D}(y', y)\mu(y', y)\,\frac{\pi_\theta(y)^\alpha}{\pi_\theta(y)^\alpha + \pi_\theta(y')^\alpha}\left(\frac{d}{d\,\pi_\theta(y)}\log \pi_\theta(y') - \frac{d}{d\,\pi_\theta(y)}\log \pi_\theta(y)\right) \\
&= \sum_{y' \in \mathcal{Y}} \tilde{\mu}(y, y')\,\frac{\pi^*(y)}{\pi^*(y) + \pi^*(y')}\,\frac{\pi_\theta(y')^\alpha}{\pi_\theta(y)^\alpha + \pi_\theta(y')^\alpha}\,\frac{1}{\pi_\theta(y)} \\
&\quad - \sum_{y' \in \mathcal{Y}} \tilde{\mu}(y, y')\,\frac{\pi^*(y')}{\pi^*(y) + \pi^*(y')}\,\frac{\pi_\theta(y)^\alpha}{\pi_\theta(y)^\alpha + \pi_\theta(y')^\alpha}\,\frac{1}{\pi_\theta(y)} \\
&= \sum_{y' \in \mathcal{Y}} \frac{\tilde{\mu}(y, y')\left(\pi^*(y)\pi_\theta(y')^\alpha - \pi^*(y')\pi_\theta(y)^\alpha\right)}{\pi_\theta(y)\left(\pi^*(y) + \pi^*(y')\right)\left(\pi_\theta(y)^\alpha + \pi_\theta(y')^\alpha\right)},
\end{aligned}
$$

$$\tag{25}$$

---

[4]Projection on the probability simplex.

where the first equality follows from (14) and by considering all the terms that include $y$ either as winner (the first sum) or loser (the second sum); the second equality is due to (23) and the fact that $\tilde{\mu}$ is symmetric. To simplify the notation, for any $y$ and $y'$, let

$$h(y, y') \overset{\text{def}}{=} \frac{\tilde{\mu}(y, y')}{\pi_\theta(y)\,\pi_\theta(y')\,\left(\pi^*(y) + \pi^*(y')\right)\left(\pi_\theta(y)^\alpha + \pi_\theta(y')^\alpha\right)}. \tag{26}$$

Then, (25) simplifies to

$$v(y) = \sum_{y' \in \mathcal{Y}} h(y, y')\,\pi_\theta(y')\left(\pi_\theta(y')^\alpha \pi^*(y) - \pi_\theta(y)^\alpha \pi^*(y')\right). \tag{27}$$

Consequently,

$$\mathbf{v}^T(\boldsymbol{\pi}_\theta - \boldsymbol{\pi}_\alpha^*) = \sum_{y \in \mathcal{Y}} v(y)\left(\pi_\theta(y) - \pi_\alpha^*(y)\right)$$

$$= \sum_{y, y' \in \mathcal{Y}} h(y, y')\left(\pi_\theta(y')^\alpha \pi^*(y) - \pi_\theta(y)^\alpha \pi^*(y')\right)\,\pi_\theta(y')\left(\pi_\theta(y) - \pi_\alpha^*(y)\right)$$

$$= \frac{1}{2}\sum_{y, y' \in \mathcal{Y}} h(y, y')\left(\pi_\theta(y')^\alpha \pi^*(y) - \pi_\theta(y)^\alpha \pi^*(y')\right)\,\pi_\theta(y')\left(\pi_\theta(y) - \pi_\alpha^*(y)\right)$$

$$+ \frac{1}{2}\sum_{y', y \in \mathcal{Y}} h(y', y)\left(\pi_\theta(y)^\alpha \pi^*(y') - \pi_\theta(y')^\alpha \pi^*(y)\right)\,\pi_\theta(y)\left(\pi_\theta(y') - \pi_\alpha^*(y')\right)$$

$$= \frac{1}{2}\sum_{y, y' \in \mathcal{Y}} h(y, y')\left(\pi_\theta(y')^\alpha \pi^*(y) - \pi_\theta(y)^\alpha \pi^*(y')\right)\,\left(\pi_\theta(y')\pi_\theta(y) - \pi_\theta(y')\pi_\alpha^*(y)\right)$$

$$+ \frac{1}{2}\sum_{y, y' \in \mathcal{Y}} h(y, y')\left(\pi_\theta(y')^\alpha \pi^*(y) - \pi_\theta(y)^\alpha \pi^*(y')\right)\,\left(\pi_\theta(y)\pi_\alpha^*(y') - \pi_\theta(y)\pi_\theta(y')\right)$$

$$= \frac{1}{2}\sum_{y, y' \in \mathcal{Y}} h(y, y')\left(\pi_\theta(y')^\alpha \pi^*(y) - \pi_\theta(y)^\alpha \pi^*(y')\right)\,\left(\pi_\theta(y)\pi_\alpha^*(y') - \pi_\theta(y')\pi_\alpha^*(y)\right)$$

$$= -\frac{z_\alpha}{2}\sum_{y, y' \in \mathcal{Y}} h(y, y')\left(\left(\pi_\theta(y')\pi_\alpha^*(y)\right)^\alpha - \left(\pi_\theta(y)\pi_\alpha^*(y')\right)^\alpha\right)\,\left(\pi_\theta(y')\pi_\alpha^*(y) - \pi_\theta(y)\pi_\alpha^*(y')\right)$$

$$= -\frac{z_\alpha}{2}\sum_{y, y' \in \mathcal{Y}} h(y, y')\left(\pi_\theta(y)\pi_\theta(y')\right)^{1+\alpha}\left(\left(\frac{\pi_\alpha^*(y)}{\pi_\theta(y)}\right)^\alpha - \left(\frac{\pi_\alpha^*(y')}{\pi_\theta(y')}\right)^\alpha\right)\,\left(\frac{\pi_\alpha^*(y)}{\pi_\theta(y)} - \frac{\pi_\alpha^*(y')}{\pi_\theta(y)}\right), \tag{28}$$

where the second equality follows from (27), the fourth equality is due to the symmetry of $h(y, y')$ with respect to $y$ and $y'$, i.e., $h(y, y') = h(y', y)$, and the sixth equality is from (20). It is easy to see that all terms in the sum in the last line are non-negative, and the sum contains at least one non-zero term if $\boldsymbol{\pi}_\theta \neq \boldsymbol{\pi}_\alpha^*$. Therefore, $\mathbf{v}^T(\boldsymbol{\pi}_\theta - \boldsymbol{\pi}_\alpha^*) < 0$ if $\boldsymbol{\pi}_\theta \neq \boldsymbol{\pi}_\alpha^*$. Consequently, $\|\boldsymbol{\pi}_\theta - \boldsymbol{\pi}_\alpha^*\|$ is strictly decreasing when moving along $\mathbf{v}$. Since both $\boldsymbol{\pi}_\theta$ and $\boldsymbol{\pi}_\alpha^*$ lie on the probability simplex, we have $\prod(\mathbf{v})^T(\boldsymbol{\pi}_\theta - \boldsymbol{\pi}_\alpha^*) \leq \mathbf{v}^T(\boldsymbol{\pi}_\theta - \boldsymbol{\pi}_\alpha^*) < 0$. It follows that for any $\boldsymbol{\pi}_\theta$ in the relative interior of the probability simplex, projection of $\mathbf{v}$ on the probability simplex is a strictly decent direction for $\|\boldsymbol{\pi}_\theta - \boldsymbol{\pi}_\alpha^*\|$.

As a result, $\pi_\alpha^*$ is the globally absorbing unique fixed point of the ODE. Furthermore, when $\mu$ is not a function of $\pi_\theta$, then $\pi_\alpha^*$ is the unique first order stationary point of the preference loss $\mathcal{L}_{\text{pref}}^{\alpha, \mu}$. In other words, $\mathcal{L}_{\text{pref}}^{\alpha, \mu}$ contains no other local mininum, local maximum, or saddle-point in the probability simplex.

# B  PROOF OF THEOREM 3

This appendix presents the proof of Theorem 3. The high-level idea, akin to Appendix A, is to show that moving along the ODE direction results in an absolute reduction of the Euclidean distance of $\pi_\theta$ from $\text{Softmax}\left(r(\cdot|x)/\alpha\right)$. The details are however substantially different from Appendix A.

We begin with the following lemma.

**Lemma 1.** *For any $\eta > 0$ and any pair of $n$-dimensional vectors $\boldsymbol{a}$ and $\boldsymbol{b}$ with positive entries, we have*

$$\sum_{i=1}^{n} \left(\frac{a_i}{b_i}\right)^{\eta} \left(\frac{b_i}{\sum_{j=1}^{n} b_j} - \frac{a_i}{\sum_{j=1}^{n} a_j}\right) \leq 0, \tag{29}$$

*and the equality holds only if $\boldsymbol{a} = c\boldsymbol{b}$ for some scalar $c$.*

*Proof of Lemma 1.* Fix an arbitrary vector $\boldsymbol{a}$ with positive entries, and consider the following function

$$f(\mathbf{x}) \stackrel{\text{def}}{=} \sum_{i=1}^{n} \left(\frac{a_i}{x_i}\right)^{\eta} \left(\frac{x_i}{\sum_{j=1}^{n} x_j} - \frac{a_i}{\sum_{j=1}^{n} a_j}\right), \qquad \text{for} \quad \mathbf{x} \in \mathbb{R}_{+}^{n}, \tag{30}$$

defined on the positive quadrant. We will show that $f(\mathbf{x}) \leq 0$, for all $x \in \mathbb{R}_{+}^{n}$. Note that if $f(\mathbf{x}) > 0$ for some $x$, then $f(c\mathbf{x}) = f(\mathbf{x})/c^{\eta} > 0$, for all $c > 0$. Therefore, without loss of generality, we confine the domain to a compact set, say to the probability simplex $\mathcal{S} \stackrel{\text{def}}{=} \{\mathbf{x} \in \mathbb{R}_{+}^{*} : \sum_{i=1}^{n} x_i = 1\}$, and show that $f(\mathbf{x}) \leq 0$ for all $\mathbf{x} \in \mathcal{S}$. In the same vein, without loss of generality we also assume that

$$\sum_{i=1}^{n} a_i = 1. \tag{31}$$

Note that $f(\mathbf{x}) = -\infty$ on the boundary of the probability simplex, that is if $x_i = 0$ for some $i$. Therefore, the maximizer $\mathbf{x}^*$ of $f$ over $\mathcal{S}$, lies in the relative interior of $\mathcal{S}$. Consequently, the gradient of the Lagrangian of $f$ at $\mathbf{x}^*$ is zero. The Lagrangian $L$ of $f$ is as follows:

$$L(\mathbf{x}, \lambda) \stackrel{\text{def}}{=} f(\mathbf{x}) + \lambda \left(\sum_{i=1}^{n} x_i - 1\right), \qquad \text{for } \mathbf{x} \in \mathcal{S}, \ \lambda \in \mathbb{R}. \tag{32}$$

Then,

$$\begin{aligned}
\frac{d}{d x_k} L(\mathbf{x}, \lambda) &= \frac{d}{d x_k} f(\mathbf{x}) + \lambda \\
&= \frac{d}{d x_k} \sum_{i=1}^{n} \left(\frac{a_i}{x_i}\right)^{\eta} \left(\frac{x_i}{\sum_{j=1}^{n} x_j} - \frac{a_i}{\sum_{j=1}^{n} a_j}\right) + \lambda \\
&= \frac{d}{d x_k} \sum_{i=1}^{n} \left(\frac{a_i^{\eta} x_i^{1-\eta}}{\sum_{j=1}^{n} x_j} - a_i^{1+\eta} x_i^{-\eta}\right) + \lambda \\
&= \frac{(1-\eta) a_k^{\eta} x_k^{-\eta}}{\sum_{j=1}^{n} x_j} - \frac{\sum_{i=1}^{n} a_i^{\eta} x_i^{1-\eta}}{\left(\sum_{j=1}^{n} x_j\right)^2} + \eta a_k^{1+\eta} x_k^{-\eta-1} + \lambda \\
&= (1-\eta) \left(\frac{a_k}{x_k}\right)^{\eta} + \eta \left(\frac{a_k}{x_k}\right)^{1+\eta} + \left[\lambda - \sum_{i=1}^{n} a_i^{\eta} x_i^{1-\eta}\right]
\end{aligned} \tag{33}$$

where the third equality is due to (31), and the last equality is because $\sum_j x_j = 1$. Consider a scalar function $h : \mathbb{R}_{+} \to \mathbb{R}_{+}$ as follows

$$h(y) \stackrel{\text{def}}{=} (1-\eta) y^{\eta} + \eta y^{1+\eta} \qquad \text{for} \quad y \geq 0. \tag{34}$$

Then, (33) simplifies to

$$\frac{d}{d x_k} L(\mathbf{x}, \lambda) = h\left(\frac{a_k}{x_k}\right) + C(\lambda, \mathbf{x}, \boldsymbol{a}), \tag{35}$$

where $C(\lambda, \mathbf{x}, \boldsymbol{a}) = \lambda - \sum_{i=1}^{n} a_i^{\eta} x_i^{1-\eta}$ is independent of $k$. Therefore, letting $\nabla_{\mathbf{x}} L(\mathbf{x}, \lambda) = 0$ at $\mathbf{x} = \mathbf{x}^*$, it follows that for any $1 \leq i < j \leq n$,

$$h\left(\frac{a_i}{x_i^*}\right) = h\left(\frac{a_j}{x_j^*}\right). \tag{36}$$

We now consider two cases for $\eta$.

**Case 1** $(\eta \leq 1)$**.** In this case, $h$ defined in (34) is an strictly increasing function. Therefore, (36) implies that $a_i/x_i^* = a_j/x_j^*$, for all $i, j \leq n$. Equivalently, $\mathbf{x}^* = c\boldsymbol{a}$ for some scalar $c > 0$. In this case, from (30), $f(\mathbf{x}^*) = 0$. The lemma then follows from the fact that $\mathbf{x}^*$ is the maximizer of $f$.

**Case 2** $(\eta > 1)$**.** In this case, $h$ is no longer increasing. In this case, $h$ is unimodal. Specifically, $h$ is strictly decreasing over $\big[0, (\eta - 1)/(\eta + 1)\big]$ and is strictly increasing over $\big[(\eta - 1)/(\eta + 1), \infty\big]$. This unimodality implies that the pre-image of any $y \in \mathbb{R}_+$ (i.e., $h^{-1}(y)$) is a set of at most two points. Consequently, (36) implies that we can partition the indices $1, \ldots, n$ into two groups $S_1$ and $S_2$ such that within each group, we have $a_i/x_i^* = a_j/x_j^*$. In other words, $a_i/x_i^* = a_j/x_j^*$ for all $(i, j) \in S_1 \times S_1$ and all $(i, j) \in S_2 \times S_2$. Equivalently, the maximum point, $\mathbf{x}^*$, belongs to the set

$$X^* \stackrel{\text{def}}{=} \big\{ \mathbf{x} \in \mathbb{R}_+^n : \ x_i = c_1 a_i \text{ for } i \leq k, \text{ and } x_i = c_2 a_i \text{ for } i > k, \text{ for some } c_1, c_2 > 0 \text{ and } k < n \big\}, \tag{37}$$

where we have assumed without loss of generality that $S_1 = \{1, \ldots, k\}$ and $S_2 = \{k + 1, \ldots, n\}$ for some $k \leq n$. We will show that $f(\mathbf{x}) \leq 0$ for all $\mathbf{x} \in X^*$.

Fix some $\mathbf{x} \in X^*$, and corresponding constants $c_1$, $c_2$, and $k$, as per (37). Let $A = \sum_{i=1}^{k} a_i$ and $B = \sum_{i=k+1}^{n} a_i$. Then,

$$
\begin{aligned}
f(\mathbf{x}) &= \sum_{i=1}^{n} \left( \frac{a_i}{x_i} \right)^{\eta} \left( \frac{x_i}{\sum_{j=1}^{n} x_j} - \frac{a_i}{\sum_{j=1}^{n} a_j} \right) \\
&= \sum_{i=1}^{n} \left( \frac{a_i}{x_i} \right)^{\eta} \left( \frac{x_i}{c_1 A + c_2 B} - \frac{a_i}{A + B} \right) \\
&= \sum_{i=1}^{k} c_1^{-\eta} \left( \frac{c_1 a_i}{c_1 A + c_2 B} - \frac{a_i}{A + B} \right) + \sum_{i=k+1}^{n} c_2^{-\eta} \left( \frac{c_2 a_i}{c_1 A + c_2 B} - \frac{a_i}{A + B} \right) \\
&= \left( \frac{c_1^{1-\eta} A}{c_1 A + c_2 B} - \frac{c_1^{-\eta} A}{A + B} \right) + \left( \frac{c_2^{1-\eta} B}{c_1 A + c_2 B} - \frac{c_2^{-\eta} B}{A + B} \right) \\
&= \frac{c_1^{1-\eta} A + c_2^{1-\eta} B}{c_1 A + c_2 B} - \frac{c_1^{-\eta} A + c_2^{-\eta} B}{A + B} \\
&= \frac{(c_1^{1-\eta} A + c_2^{1-\eta} B)(A + B) - (c_1^{-\eta} A + c_2^{-\eta} B)(c_1 A + c_2 B)}{(c_1 A + c_2 B)(A + B)} \\
&= \frac{(c_1 - c_2)(c_1^{-\eta} - c_2^{-\eta}) A B}{(c_1 A + c_2 B)(A + B)} \\
&\leq 0,
\end{aligned}
$$

and the inequality in the last line holds with equality iff either $A$ or $B$ are zero (note that $c_1, c_2, \eta > 0$), which is the case only if $\mathbf{x} = c_1 \boldsymbol{a}$ or $\mathbf{x} = c_2 \boldsymbol{a}$. The lemma then follows from the fact that $\mathbf{x}^*$ is the maximizer of $f$.

This completes the proof of Lemma 1. $\qquad\square$

We proceed with the proof of the theorem. Given the rewards $r(\cdot|\cdot)$ in the $n$-ary BT model (see Section 5), let

$$\pi^*(\cdot|\cdot) \stackrel{\text{def}}{=} \text{Softmax}\big(r(\cdot|\cdot)\big). \tag{38}$$

For any $(x; y_1, \ldots, y_n) \in \mathcal{X} \times \mathcal{Y}^n$, let

$$\bar{\mathcal{D}}(x; y_1, \ldots, y_n) \stackrel{\text{def}}{=} \frac{\sum_{i=1}^{n} \mathcal{D}(x; y_1, \ldots, y_n; i)}{\sum_{i=1}^{n} \pi^*(y_i|x)}. \tag{39}$$

It then follows from the consistency of $\mathcal{D}$ with the $n$-ary BT model that for any $(x; y_1, \ldots, y_n) \in \mathcal{X} \times \mathcal{Y}^n$ and $i = 1, \ldots, n$

$$\mathcal{D}(x; y_1, \ldots, y_n; i) = \bar{\mathcal{D}}(x; y_1, \ldots, y_n) \, \pi^*(y_i|x). \tag{40}$$

We further define

$$\tilde{\mathcal{D}}(x; [y]) \stackrel{\text{def}}{=} \bar{\mathcal{D}}(x; y_1, \ldots, y_n)\, \mu(x; y_1, \ldots, y_n). \tag{41}$$

For brevity of notation, we denote $y_1, \ldots, y_n$ by $[y]$ and denote $1, \ldots, n$ by $[n]$. The loss function $\mathcal{L}_{\text{pref-}n}^{\alpha,\mu}(\pi, \mathcal{D})$ defined in (16) can then be simplified to

$$
\begin{aligned}
\mathcal{L}_{\text{pref-}n}^{\alpha,\mu}(\pi, \mathcal{D}) &= -\frac{1}{\alpha} \mathbb{E}_{(x; y_1, \ldots, y_n; i^*)\sim\mathcal{D}} \left[ \mu(y_1, \ldots, y_n \mid x) \log \frac{\pi(y_{i^*} \mid x)^\alpha}{\sum_{i=1}^n \pi(y_i \mid x)^\alpha} \right] \\
&= -\frac{1}{\alpha} \sum_{(x; [y]; i^*)\in\mathcal{X}\times\mathcal{Y}^n\times[n]} \mathcal{D}(x; [y]; i^*)\, \mu([y]|x) \log \frac{\pi(y_{i^*} \mid x)^\alpha}{\sum_{i=1}^n \pi(y_i \mid x)^\alpha} \\
&= -\frac{1}{\alpha} \sum_{(x; [y]; i^*)\in\mathcal{X}\times\mathcal{Y}^n\times[n]} \tilde{D}(x; [y])\, \pi^*(y_{i^*}|x) \log \frac{\pi(y_{i^*} \mid x)^\alpha}{\sum_{i=1}^n \pi(y_i \mid x)^\alpha} \\
&= -\frac{1}{\alpha} \sum_{(x; [y])\in\mathcal{X}\times\mathcal{Y}^n} \tilde{D}(x; [y]) \sum_{i=1}^n \pi^*(y_i|x) \log \frac{\pi(y_i \mid x)^\alpha}{\sum_{j=1}^n \pi(y_j \mid x)^\alpha},
\end{aligned}
$$

where the third equality is due to (40) and (41).

In the rest of the proof, without loss of generality, we consider a single fixed $x \in \mathcal{X}$, and remove $x$ from the notations for the sake of notation brevity. Let

$$\pi_\alpha^*(\cdot) \stackrel{\text{def}}{=} \text{Softmax}\left(r(\cdot)/\alpha\right). \tag{42}$$

It follows that for any $y \in \mathcal{Y}$,

$$\pi_\alpha^*(y) = \frac{\pi^*(y)^{1/\alpha}}{\sum_{\tilde{y}\in\mathcal{Y}} \pi^*(\tilde{y})^{1/\alpha}}. \tag{43}$$

Let $\boldsymbol{\pi}$ and $\boldsymbol{\pi}_\alpha^*$ be the vector representation of $\pi(y)$ and $\pi_\alpha^*(y)$ for all $y \in \mathcal{Y}$. Then, for $\mathbf{v} \stackrel{\text{def}}{=} -\nabla_\pi \mathcal{L}_{\text{pref-}n}^{\alpha,\mu}(\pi, \mathcal{D})$ we have

$$
\begin{aligned}
\left(\boldsymbol{\pi} - \boldsymbol{\pi}^{*1/\alpha}\right)^T \mathbf{v} &= -\left(\boldsymbol{\pi} - \boldsymbol{\pi}_\alpha^*\right)^T \nabla_\pi \mathcal{L}_{\text{pref-}n}^{\alpha,\mu}(\pi, \mathcal{D}) \\
&= \frac{1}{\alpha} \sum_{[y]\in\mathcal{Y}^n} \tilde{D}([y])\, \left(\boldsymbol{\pi} - \boldsymbol{\pi}_\alpha^*\right)^T \nabla_\pi \sum_{i=1}^n \pi^*(y_i) \log \frac{\pi(y_i \mid x)^\alpha}{\sum_{j=1}^n \pi(y_j \mid x)^\alpha} \\
&= \frac{1}{\alpha} \sum_{[y]\in\mathcal{Y}^n} \tilde{D}([y]) \sum_{\tilde{y}\in\mathcal{Y}} (\pi(\tilde{y}) - \pi_\alpha^*(\tilde{y})) \frac{d}{d\tilde{y}} \sum_{i=1}^n \pi^*(y_i) \log \frac{\pi(y_i \mid x)^\alpha}{\sum_{j=1}^n \pi(y_j \mid x)^\alpha} \\
&= \frac{1}{\alpha} \sum_{[y]\in\mathcal{Y}^n} \tilde{D}([y]) \sum_{k=1}^n (\pi(y_k) - \pi_\alpha^*(y_k)) \frac{d}{d y_k} \sum_{i=1}^n \pi^*(y_i) \log \frac{\pi(y_i \mid x)^\alpha}{\sum_{j=1}^n \pi(y_j \mid x)^\alpha}.
\end{aligned} \tag{44}
$$

For any $[y] = (y_1, \ldots, \mathbf{y}_n) \in \mathcal{Y}^n$, let

$$A([y]) \stackrel{\text{def}}{=} \frac{1}{\alpha} \sum_{k=1}^n (\pi(y_k) - \pi_\alpha^*(y_k)) \frac{d}{d y_k} \sum_{i=1}^n \pi^*(y_i) \log \frac{\pi(y_i \mid x)^\alpha}{\sum_{j=1}^n \pi(y_j \mid x)^\alpha}. \tag{45}$$

It then follows from (44) that:

$$\mathbf{v}^T \left(\boldsymbol{\pi} - \boldsymbol{\pi}^{*1/\alpha}\right) = \sum_{[y]\in\mathcal{Y}^n} \tilde{D}([y])\, A([y]). \tag{46}$$

We proceed to compute $A([y])$. For $k = 1, \ldots, n$,

$$
\begin{aligned}
\frac{d}{d y_k} \sum_{i=1}^n \frac{\pi(y_i \mid x)^\alpha}{\sum_{j=1}^n \pi(y_j \mid x)^\alpha} &= \frac{d}{d y_k} \sum_{i=1}^n \pi^*(y_i) \left( \log \pi(y_i)^\alpha - \log \sum_{j=1}^n \pi(y_j)^\alpha \right) \\
&= \alpha \frac{\pi^*(y_k)}{\pi(y_k)} - \left( \sum_{i=1}^n \pi^*(y_i) \right) \frac{d}{d y_k} \log \sum_{j=1}^n \pi(y_j)^\alpha \\
&= \alpha \frac{\pi^*(y_k)}{\pi(y_k)} - \alpha \left( \sum_{i=1}^n \pi^*(y_i) \right) \frac{\pi(y_k)^{\alpha-1}}{\sum_{j=1}^n \pi(y_j)^\alpha}
\end{aligned} \tag{47}
$$

Plugging this into the definition of $A\big([y]\big)$ in (45), we obtain

$$
\begin{aligned}
A\big([y]\big) &= \sum_{k=1}^{n} (\pi(y_k) - \pi_\alpha^*(y_k)) \left( \frac{\pi^*(y_k)}{\pi(y_k)} - \left( \sum_{i=1}^{n} \pi^*(y_i) \right) \frac{\pi(y_k)^{\alpha-1}}{\sum_{j=1}^{n} \pi(y_j)^\alpha} \right) \\
&= \sum_{k=1}^{n} \pi(y_k) \left( \frac{\pi^*(y_k)}{\pi(y_k)} - \left( \sum_{i=1}^{n} \pi^*(y_i) \right) \frac{\pi(y_k)^{\alpha-1}}{\sum_{j=1}^{n} \pi(y_j)^\alpha} \right) \\
&\quad + \sum_{k=1}^{n} \pi_\alpha^*(y_k) \left( \left( \sum_{i=1}^{n} \pi^*(y_i) \right) \frac{\pi(y_k)^{\alpha-1}}{\sum_{j=1}^{n} \pi(y_j)^\alpha} - \frac{\pi^*(y_k)}{\pi(y_k)} \right) \\
&= \sum_{k=1}^{n} \pi^*(y_k) - \left( \sum_{i=1}^{n} \pi^*(y_i) \right) \frac{\sum_{k=1}^{n} \pi(y_k)^\alpha}{\sum_{j=1}^{n} \pi(y_j)^\alpha} \\
&\quad + \sum_{k=1}^{n} \pi_\alpha^*(y_k) \left( \left( \sum_{i=1}^{n} \pi^*(y_i) \right) \frac{\pi(y_k)^{\alpha-1}}{\sum_{j=1}^{n} \pi(y_j)^\alpha} - \frac{\pi^*(y_k)}{\pi(y_k)} \right) \\
&= \sum_{k=1}^{n} \pi_\alpha^*(y_k) \left( \left( \sum_{i=1}^{n} \pi^*(y_i) \right) \frac{\pi(y_k)^{\alpha-1}}{\sum_{j=1}^{n} \pi(y_j)^\alpha} - \frac{\pi^*(y_k)}{\pi(y_k)} \right) \\
&= \frac{\sum_{i=1}^{n} \pi^*(y_i)}{\sum_{i=1}^{n} \pi^*(y_i)^{1/\alpha}} \sum_{k=1}^{n} \left( \frac{\pi^*(y_k)}{\pi(y_k)^\alpha} \right)^{1/\alpha} \left( \frac{\pi(y_k)^\alpha}{\sum_{j=1}^{n} \pi(y_j)^\alpha} - \frac{\pi^*(y_k)}{\sum_{i=1}^{n} \pi^*(y_i)} \right) \\
&\leq 0 \qquad (\texttt{"="} \text{ only if } \boldsymbol{\pi}^* = c\boldsymbol{\pi} \text{ for some scalar } c > 0),
\end{aligned}
\tag{48}
$$

where the last equality is due to (43), and the inequality in the last line follows from Lemma 1 by letting $a_k = \pi^*(y_k)$, $b_k = \pi(y_k)^\alpha$, and $\eta = 1/\alpha$. Plugging this into (46), it follows that

$$
- \left( \nabla_\pi \mathcal{L}_{\text{pref-}n}^{\alpha,\mu}(\pi, \mathcal{D}) \right)^T (\boldsymbol{\pi} - \boldsymbol{\pi}_\alpha^*) = \mathbf{v}^T (\boldsymbol{\pi} - \boldsymbol{\pi}_\alpha^*) < 0
\tag{49}
$$

if $\boldsymbol{\pi} \neq \boldsymbol{\pi}_\alpha^*$. Consequently, $\|\boldsymbol{\pi} - \boldsymbol{\pi}_\alpha^*\|$ is strictly decreasing when moving along $\mathbf{v}$. Since both $\boldsymbol{\pi}$ and $\boldsymbol{\pi}_\alpha^*$ lie on the probability simplex, we have $\prod(\mathbf{v})^T (\boldsymbol{\pi} - \boldsymbol{\pi}_\alpha^*) \leq \mathbf{v}^T (\boldsymbol{\pi} - \boldsymbol{\pi}_\alpha^*) < 0$. It follows that for any $\boldsymbol{\pi}$ in the relative interior of the probability simplex, projection of $\mathbf{v}$ on the probability simplex is a strictly decent direction for $\|\boldsymbol{\pi} - \boldsymbol{\pi}_\alpha^*\|$. As a result, $\boldsymbol{\pi}_\alpha^*$ is the globally absorbing unique fixed point of the ODE. This completes the proof of Theorem 3.

## C  PROOF OF THEOREM 4

Here we present the proof of Theorem 4. The high level idea is to show that $\mathcal{L}_{\text{rank}}^{\alpha,[\mu]}(\pi, \mathcal{D})$ can be equivalently written as the sum of $\mathcal{L}_{\text{pref-}n}^{\alpha,\mu_k}(\pi, \mathcal{D}_k)$ for appropriately defined $\mathcal{D}_k$, for $k = 1, \ldots, n-1$; where each $\mathcal{D}_k$ is consistent with the $(n-k+1)$-ary BT model (defined in Section 5). We then use Theorem 3, and in particular (49) in the proof of Theorem 3, to conclude that the softmax distribution is a globally absorbing fixed point of $-\nabla \mathcal{L}_{\text{pref-}n}^{\alpha,\mu_k}(\pi, \mathcal{D}_k)$ for $k = 1, \ldots, n-1$, and is therefore a globally absorbing fixed point of their sum, $-\nabla \mathcal{L}_{\text{rank}}^{\alpha,[\mu]}(\pi, \mathcal{D})$.

As in the previous appendices, without loss of generality we prove the theorem for a single fixed $x \in \mathcal{X}$, and remove $x$ from the equations for notation brevity. To further simplify the notation, without loss of generality, we also remove the permutation $\tau$ from the equations, and represent the ranking by mere order of the indices, that is we assume that $y_1 \succ y_2 \succ \cdots \succ y_n$. With these new conventions, the ranking loss (17) simplifies to

$$
\mathcal{L}_{\text{rank}}^{\alpha,[\mu]}(\pi, \mathcal{D}) \overset{\text{def}}{=} -\frac{1}{\alpha} \mathbb{E}_{(y_1,\ldots,y_n)\sim\mathcal{D}} \left[ \sum_{k=1}^{n-1} \mu_k(y_k,\ldots,y_n) \log \frac{\pi(y_k)^\alpha}{\sum_{j=k}^{n} \pi(y_j)^\alpha} \right].
\tag{50}
$$

For $k = 1, \ldots, n-1$, we define an $(n-k+1)$-ary preference distribution $\mathcal{D}_k$ as follows. For any $(y_1, \ldots, y_{n-k+1}) \in \mathcal{Y}^n$ and $i = 1, \ldots, n-k+1$,

$$\mathcal{D}_k(y_1, \ldots, y_{n-k+1}; i) = \frac{1}{(n-k)!} \sum_{\substack{(z_1, \ldots, z_{k-1}) \in \mathcal{Y}^{k-1} \\ \text{Permutation } \tau: (1, \ldots, n-k) \to (1, \ldots, i, \ldots, n-k+1)}} \mathcal{D}(z_1, \ldots, z_{k-1}, y_i, y_{\tau(1)}, \ldots, y_{\tau(n-k)}).$$

(51)

From (50), we have

$$\mathcal{L}_{\text{rank}}^{\alpha, [\mu]}(\pi, \mathcal{D}) = -\frac{1}{\alpha} \mathbb{E}_{(y_1, \ldots, y_n) \sim \mathcal{D}} \left[ \sum_{k=1}^{n-1} \mu_k(y_k, \ldots, y_n) \log \frac{\pi(y_k)^\alpha}{\sum_{j=k}^n \pi(y_j)^\alpha} \right]$$

$$= -\frac{1}{\alpha} \sum_{k=1}^{n-1} \mathbb{E}_{(y_1, \ldots, y_n) \sim \mathcal{D}} \left[ \mu_k(y_k, \ldots, y_n) \log \frac{\pi(y_k)^\alpha}{\sum_{j=k}^n \pi(y_j)^\alpha} \right]$$

$$= -\frac{1}{\alpha} \sum_{k=1}^{n-1} \sum_{(y_1, \ldots, y_n) \in \mathcal{Y}^n} \mathcal{D}(y_1, \ldots, y_n) \left[ \mu_k(y_k, \ldots, y_n) \log \frac{\pi(y_k)^\alpha}{\sum_{j=k}^n \pi(y_j)^\alpha} \right]$$

$$= -\frac{1}{\alpha} \sum_{k=1}^{n-1} \sum_{y_k, \ldots, y_n} \sum_{(y_1, \ldots, y_{k-1}) \in \mathcal{Y}^{k-1}} \mathcal{D}(y_1, \ldots, y_n) \left[ \mu_k(y_k, \ldots, y_n) \log \frac{\pi(y_k)^\alpha}{\sum_{j=k}^n \pi(y_j)^\alpha} \right]$$

$$= -\frac{1}{\alpha} \sum_{k=1}^{n-1} \sum_{y_k, \ldots, y_n} \frac{\mathcal{D}_k(y_1, \ldots, y_{n-k+1}; 1)}{\left( (n-k)! \right)^2} \left[ \mu_k(y_k, \ldots, y_n) \log \frac{\pi(y_k)^\alpha}{\sum_{j=k}^n \pi(y_j)^\alpha} \right]$$

$$= -\frac{1}{\alpha} \sum_{k=1}^{n-1} \frac{1}{(n-k+1)! \, (n-k)!} \mathbb{E}_{(y_k, \ldots, y_n; i) \sim \mathcal{D}_k} \left[ \mu_k(y_k, \ldots, y_n) \log \frac{\pi(y_k)^\alpha}{\sum_{j=k}^n \pi(y_j)^\alpha} \right]$$

$$= \sum_{k=1}^{n-1} \frac{\mathcal{L}_{\text{pref-}n}^{\alpha, \mu_k}(\pi, \mathcal{D}_k)}{(n-k+1)! \, (n-k)!}.$$

Let $\boldsymbol{\pi}$ and $\boldsymbol{\pi}_\alpha^*$ be the vector representations of $\pi$ and the softmax distribution $\pi_\alpha^*$ (defined in (43)), and $\mathbf{v} \stackrel{\text{def}}{=} -\nabla_\pi \mathcal{L}_{\text{rank}}^{\alpha, [\mu]}(\pi, \mathcal{D})$. Then,

$$(\boldsymbol{\pi} - \boldsymbol{\pi}_\alpha^*)^T \mathbf{v} = -(\boldsymbol{\pi} - \boldsymbol{\pi}_\alpha^*)^T \nabla \sum_{k=1}^{n-1} \frac{\mathcal{L}_{\text{pref-}n}^{\alpha, \mu_k}(\pi, \mathcal{D}_k)}{(n-k+1)! \, (n-k)!}$$

$$= \sum_{k=1}^{n-1} \frac{-(\boldsymbol{\pi} - \boldsymbol{\pi}_\alpha^*)^T \nabla \mathcal{L}_{\text{pref-}n}^{\alpha, \mu_k}(\pi, \mathcal{D}_k)}{(n-k+1)! \, (n-k)!}$$

$$\leq 0,$$

where the last inequality follows from (49), and it holds with equality only if $\boldsymbol{\pi} \neq \boldsymbol{\pi}_\alpha^*$. Since both $\boldsymbol{\pi}$ and $\boldsymbol{\pi}_\alpha^*$ lie on the probability simplex, we have $\prod(\mathbf{v})^T (\boldsymbol{\pi} - \boldsymbol{\pi}_\alpha^*) \leq \mathbf{v}^T (\boldsymbol{\pi} - \boldsymbol{\pi}_\alpha^*) < 0$. Then, following a similar argumnet as in the last paragraph of Appendix B, we conclude that $\boldsymbol{\pi}_\alpha^*$ is the globally absorbing unique fixed point of the ODE. This completes the proof of Theorem 4.

# D EXPERIMENT DETAILS

## D.1 DETAILS FOR THE ALPACAFARM EXPERIMENT OF SECTION 7.1

We performed the alignment procedure on 4 NVIDIA H100 (94 GiB) GPUs. For each method, we trained the model for four epochs and reported the maximum win-rate against the SFT model. The training batch size per GPU device was set to one and the gradient accumulation step was 16. The range of hyper-parameters considered for each method is given as follows: R-DPO: $\beta \in \{0.01, 0.1\}$, $\alpha \in \{0.001, 0.01, 0.1\}$, CPO: $\beta \in \{0.001, 0.01, 0.1\}$, $\alpha \in \{0.0001, 0.001, 0.01\}$, SimPO: $\beta \in \{2, 2.5\}$, $\gamma \in \{1, 1.5\}$ (the suggested range in SimPO), KTO: $\beta \in \{0.01, 0.1\}$, $\lambda_D = \lambda_U = 1$, IPO: $\tau \in \{0.001, 0.01, 0.1\}$, DPO: $\beta \in \{0.001, 0.01, 0.1\}$, and SPO: $\alpha \in \{0.001, 0.01\}$, $\beta \in \{0.001, 0.01\}$. Table 5 shows the sensitivity of SPO to its hyperparameters.

|  | $\alpha = 0.01$ | $\alpha = 0.001$ |
|---|---|---|
| $\beta = 0.01$ | **60.83** | 59.82 |
| $\beta = 0.001$ | 57.15 | 59.99 |

Table 5: SPO win-rate for different $\alpha$ and $\beta$ values.

### D.2 DETAILS FOR THE EXPERIMENT OF SECTION 7.2

We trained the models on NVIDIA A100 (40 GiB) GPUs. We used a batch size of 32 samples (each containing four responses) for all algorithms. The reference model in all algorithms was identical to the SFT model. All alignment loss functions were optimized using AdamW with 5,000 warm-up iterations.

For SPO, we trained both basic (i.e., $\gamma = 0$) and weighted version. For weighted SPO, we set $\gamma = 0.01$ without sweeping, and in the ranking experiment we used decayed weight functions $\eta^k \mu_k$ for $\eta \in 1, 0.5$, see (17). For other SPO parameters, we swept over $\beta \in \{0.01, 0.1\}$, and $\alpha \in \{0.001\}$. For $\mathcal{D}_{\mathrm{KL}}$ computation, we used intermittent batch generation of samples, generating a batch of 32 samples from $\pi_\theta$ once every 8 iterations (i.e., $T = 8$). For other algorithms, we swept over the following sets of hyperparameters: DPO: $\beta \in \{0.0001, 0.001, 0.01\}$, S-DPO for ranking: $\beta \in \{0.0001, 0.001, 0.01\}$, and S-DPO for best-of-$n$: $\beta \in \{0.0001, 0.001, 0.01\}$. For For training the S-DPO algorithm on the best-of-$n$ dataset, we consider each top-rank response as the positive response and the corresponding lower-rank responses as the corresponding negative set of responses. For training S-DPO on ranking dataset, each of the 1st, 2nd, and 3rd rank responses serve as positive responses with the corresponding negative set containing the corresponding lower rank responses.

We computed the win rates of all methods against the best-of-$n$ SFT model using GPT4o-2024-08-06 once every 1000 iterations, and reported the peak win-rate for each method. Each win-rate was averaged over 1,000 story-pair instances, resulting in an estimation error with standard deviation smaller than 0.015.

## E DETAILS OF GENERATION OF RANKING DATASET FOR THE TINYSTORIES EXPERIMENT

We created a preference dataset to align stories with older age groups. Specifically, for each pair of stories generated by the reference model, we asked GPT4o–2024-08-06 to evaluate them based on clarity and coherence, writing quality, and whether they are interesting and engaging for high school students. The API was asked to evaluate each story independently based on these criteria, and identify its strengths and weaknesses compared to other stories; and then suggest a ranking of stories from best to worst. The prompt used for generating the dataset is provided at the end of this subsection.

We generated a set of 100,000 stories independently from the 110M-parameter pre-trained model (Karpathy, 2023), and grouped them into a set of 25,000 samples each containing four stories. To enhance the quality of the ranking dataset, for each sample we used the prompt to rank the stories twice, reversing the order of the stories in the second evaluation. We retained samples only if both evaluations showed a consistent ranking. After this filtration, 5,000 samples remained for use in the ranking dataset. The dataset is available online at (Anonymous, 2024).

Prompt for Generation of the Ranking Dataset for TinyStories Experiment:

```
You are tasked with deciding which of the four short stories below,
written by high school students, is better suited for publication in the
high school newspaper.

**Story 1:**   {}

**Story 2:**   {}

**Story 3:**   {}

**Story 4:**   {}
```

**Your Task:**
```
1.  **Evaluate Each Story Individually:**
 Identify the **strengths** and **weaknesses** of each story, focusing on:
   - **Engagement:** Is the story interesting and likely to captivate high
school students?
   - **Clarity and Coherence:** Is the story well-organized and easy to
follow?
   - **Writing Quality:** Assess the grammar, vocabulary, and overall
language use.

2.  **Make a Final Decision:**
   - Based on your evaluations, decide which story is better suited for
publication.  Rank the stories from best to worst.
```

**Response Format:**
```
***
**Evaluation:**
**Story 1:**
- **Strengths compared to other stories:**
   - [List strengths]
- **Weaknesses compared to other stories:**
   - [List weaknesses]
--
**Story 2:**
- **Strengths compared to other stories:**
   - [List strengths]
- **Weaknesses compared to other stories:**
   - [List weaknesses]
--
**Story 3:**
- **Strengths compared to other stories:**
   - [List strengths]
- **Weaknesses compared to other stories:**
   - [List weaknesses]
--
**Story 4:**
- **Strengths compared to other stories:**
   - [List strengths]
- **Weaknesses compared to other stories:**
   - [List weaknesses]
--
**Conclusion:**
- **Overall Ranking of the Stories:**
   1.  **1st Place:** Story [1, 2, 3, or 4]
   2.  **2nd Place:** Story [1, 2, 3, or 4]
   3.  **3rd Place:** Story [1, 2, 3, or 4]
   4.  **4th Place:** Story [1, 2, 3, or 4]
***
```

**Guidelines:**
```
   - In each evaluation, compare the story to the others, noting unique
strengths and weaknesses.
   - Evaluation of each story shold not be influenced from prior
evaluations that you provided earlier.
   - Do not let the presentation order affect your judgment; treat all
stories equally.
```

