# OpenReview forum: "Soft Preference Optimization:  Aligning Language Models to Expert Distributions"
_ICLR.cc/2025/Conference — Submitted to ICLR 2025_

### Official Review · Reviewer_nmc9 · 2024-10-24

**Soundness:** 3
**Presentation:** 2
**Contribution:** 2
**Rating:** 6
**Confidence:** 4

**Summary:**

The authors propose Soft Preference Optimization (SPO), which optimizes two key components: (1) a preference loss function aimed at maximizing the likelihood of favoring the chosen response over the rejected one, and (2) a KL-divergence regularization term applied to the model's samples. The authors extend SPO to handle other types of preferences, such as best-of-n and ranked preferences. They conduct experiments using various types of preference data, including pairwise, best-of-n, and ranked preferences. The ablation study further validates the choice of the regularization term and the additional weight function used in the objective.

**Strengths:**

The approach of integrating preference loss with explicit regularization makes sense. The experimental results across different preference settings appear promising.

**Weaknesses:**

1. While the authors claim that reward-model-free alignment methods offer advantages, these methods, which solely optimize a preference dataset, may overfit or overestimate out-of-distribution data. This can result in degraded performance during inference. Therefore, to convincingly demonstrate the effectiveness of the proposed method, it is crucial to compare it against RLHF approaches like PPO. This can be achieved by replacing the preference loss with an on-policy loss term for a fair comparison.

2. The derivation of Theorem 1, as well as the subsequent theorems, does not account for the regularization term, which is typically present in the analytic solutions of PPO or DPO. This omission compromises the theoretical results, given that the proposed method essentially optimizes a combination of two loss terms.

3. The design of $\mu$ lacks intuition despite its complication, e.g., subtracting the average in a batch. Also the effect of $\gamma$ would be clearer with an ablation study of sweeping this hyper-parameter.

4. Missing data on the top of page 9: "improved win-rate from xx% to xx%".

5. Missing reference to related alignment methods [1,2,3] that match the model's output distribution.

[1] Sessa, Pier Giuseppe, et al. "Bond: Aligning llms with best-of-n distillation." Arxiv (2024).

[2] Gui, Lin, Cristina Gârbacea, and Victor Veitch. "BoNBoN Alignment for Large Language Models and the Sweetness of Best-of-n Sampling." Arxiv (2024).

[3] Ji, Haozhe, et al. "Towards efficient and exact optimization of language model alignment." ICML (2024).

**Questions:**

1. Could the authors provide a more detailed empirical comparison between PPO and SPO? Since SPO requires on-policy data sampling to compute the KL-regularization, introducing additional complexity that is typically avoided by off-policy alignment methods. Thereby, the key difference in performance between PPO and SPO might stem from the first term in Eq.(1). Would optimizing on off-policy preference data offer a significant advantage over on-policy sampled data, as in PPO, and could this be further explored? For example, training both methods on the same preference dataset, then evaluating them on a held-out test set (in-distribution) and a separate dataset from a different domain (out-of-distribution) respectively.

2. Could the authors provide a clearer interpretation of the solution obtained by SPO when regularization is applied under the BT assumption and show how the regularizer affects the convergence properties of the algorithm, as this represents the actual objective being optimized.

3. Could the authors conduct a more thorough analysis of the weighting function by breaking down its individual components and performing an ablation study, as it would help to justify the necessity of its complex form.

---

> ### Author Response · Authors · 2024-11-28
> **Authors Response 1/2**
>
> Thank you for your insightful comments. In the revision,  we addressed the issues raised in your comments, which resulted in a significant improvement of the paper.
>
> > Question 1. Could the authors provide a more detailed empirical comparison between PPO and SPO? Since SPO requires on-policy data sampling to compute the KL-regularization, introducing additional complexity that is typically avoided by off-policy alignment methods. Thereby, the key difference in performance between PPO and SPO might stem from the first term in Eq.(1). Would optimizing on off-policy preference data offer a significant advantage over on-policy sampled data, as in PPO, and could this be further explored? For example, training both methods on the same preference dataset, then evaluating them on a held-out test set (in-distribution) and a separate dataset from a different domain (out-of-distribution) respectively.
>
>
>  **On-Policy vs. Off-Policy in PPO and SPO:**
>
>  Both PPO and SPO learn off-policy when using a preference dataset, as such datasets are typically generated by a different behavior policy than $\pi_\theta$. In PPO, while policy optimization is on-policy, the reward modeling phase is typically off-policy. In SPO, the KL-regularization term is computed on-policy, but preference alignment is performed off-policy. Given that both methods mix on-policy and off-policy elements, there is no inherent advantage or disadvantage in this regard.
>
> **A key difference between PPO and SPO:**
>
> From an RL perspective, the difference between PPO and SPO resembles the distinction between actor-critic methods and policy-gradient methods. PPO employs a reward model (critic) to update the policy network (actor), whereas SPO directly updates the policy based on preference data (akin to direct policy-gradient), bypassing the need for a reward model. This direct approach simplifies the pipeline and avoids potential bias transfer from a reward model.
>
> **Empirical Comparison:**
>
> We did not include PPO in our experiments because reward-free models like DPO reportedly outperform PPO. Therefore, we prioritized comparisons against state-of-the-art methods, and since SPO outperforms these reward-free methods, we assumed it would also outperform PPO. We acknowledge the importance of a detailed empirical comparison between PPO and SPO. Due to resource constraints, we were unable to include this comparison in the current version, because our resources are used up by the new Llama3 experiments in the revision.  However, we recognize its value and plan to conduct these experiments for the final version of the paper. Please also note that our evaluations already include assessments on held-out test sets, addressing overfitting concerns.
>
> > Question 2: Could the authors provide a clearer interpretation of the solution obtained by SPO when regularization is applied under the BT assumption and show how the regularizer affects the convergence properties of the algorithm, as this represents the actual objective being optimized.
>
>
> **Asymptotic Regime and Regularization**
>
>    Our current theorems focus on the asymptotic regime where the size of the preference dataset approaches infinity. In this setting, regularization becomes unnecessary for any alignment method, including SPO. Therefore, setting the regularization parameter $\beta = 0$ is optimal. A key advantage of SPO in this regime is that it converges to the expert model, while methods like DPO and RLHF tend to converge to deterministic policies.
>
> **Challenges in Deriving Analytical Solutions with Regularization**
>
> We acknowledge the importance of understanding the impact of regularization when $\beta > 0$. Note that Dkl Regularization primarily helps prevent distributional shift outside the dataset. Such distribution shift howeve does not occur in  tabular model, becase tabular model has a separate exclusive parameter dedicated to the probability of each response. This makes the study of $\beta$'s impact less relevant in this simplified context. On the other hand, a theoretical result would be very challenging for non-tabular models.
>
>
> Extending theoretical results to include the regularization term for $\beta > 0$ is challenging, even in the tabular model. Even in the tabular model with asymptotically large datasets and additional simplifying assumptions (specific $\mu$ and sample distributions, and setting $\alpha = \beta = 1$), the analytical solution reduces to solving an equation of the form: $\pi_{\theta}(y) \log \pi_{\theta}(y) + a \pi_{\theta}(y) - b = 0,$ where $a$ and $b$ depend on $\pi_{\text{expert}}$ and $\pi_{\text{ref}}$, respectively. Involving the term $x \log x$, this equation lacks a closed-form solution using standard algebraic operations or transcendental functions. Therefore, deriving an explicit analytical solution for SPO when $\beta > 0$ is unlikely.

---

> ### Author Response · Authors · 2024-11-28
> **Authors Response 2/2**
>
> > Question 3. Could the authors conduct a more thorough analysis of the weighting function by breaking down its individual components and performing an ablation study, as it would help to justify the necessity of its complex form.
>
> Thanks for pointing the necessity of further intuition for the $\mu$ function. Eq. (13) of the paper defines $\mu$ as:
>
> $\mu(y_1,y_2) = 2\sigma\Big(\big(\pi_\theta(y_1)+\pi_\theta(y_2)\big)^{\gamma}-\hat{E} \big[\big(\pi_\theta(y_1')+\pi_\theta(y_2')\big)^{\gamma}\big]\Big).$
>
> The intuition is to reduce the weight of a pair $(y_1,y_2)$ if both responses $y_1$ and $y_2$ are weak in the sense that thy have low probability under $\pi_\theta$. This $\mu$ aims to improve the alignment quality by focussing less on discreminating between two *likely-bad* responses, while focussing more on getting prefereces correctly for *(likely good, likely bad)* and *(likely good,likely good)* pairs. This is satisfied though the additive expresson $\pi_\theta(y_1)+\pi_\theta(y_2)$ in the definition of $\mu$.
>
> The significance of a pair is then measured in relation to the average significance of the all pairs in the batch; hence the term $-\hat{E}\big[\big(\pi_\theta(y_1')+\pi_\theta(y_2')\big)^{\gamma}\big]$ in the definition of $\mu$. Moreover, since probability of a response typicly shrinks exponentially with the response length, we used an exponent $\gamma$ (which was set to 0.01 in all out experiments) to prevent $\mu$ from being dominated by a single pair.
>
> Note that $\mu$ boils down to the uniform distribution if all pairs in the batch have similar probabilities. On the other hand, if the probability of a pair is much smaller than the average probability of the pairs in the batch, then the $\mu$ function would assign a close to zero weight to this pair.
>
>
> Also note that the above $\mu$ is presented only as an example, and it is likely that there exist better forms for the weight function, development and evaluation of which we differ to future works. Significantly, Theorem 2 provides convergence guarantees in the asymptotic case for any symmetric $\mu$ not necessarily limited to this specific form.
>
> We discuss the above intuitions in the revised manuscript.
>
>
> > Missing data on page 9, and references to related works.
>
> Thank you for bringing this to our attention; we have corrected the missing data on page 9 and added the references to the related works section.

---

> > ### Comment · Reviewer_nmc9 · 2024-12-01
> >
> > Thanks for the author's response, I am not convinced by the speculative conclusion that "SPO may outperform PPO because DPO is reported to outperform PPO" as the real performance of PPO depends heavily on the correct implementation. Also, PPO is not an off-policy algorithm and we only use off/on-policy to describe the policy optimization algorithm but not the reward modeling phase. You can say the learned reward model may overestimate out-of-domain samples, but this problem is also encountered by reward-free policy optimization algorithms as the policy is only trained on in-domain data generated by some unknown policy, and the performance at inference time is not guaranteed.

---

> > > ### Author Response · Authors · 2024-12-02
> > >
> > > Thanks for the reviewer for their feedback.
> > >
> > > > Experiment on PPO
> > >
> > > We prioritized conducting an experiment to test PPO on Llama2-7B, using the implementation in AlpacaFarm codebase for both reward-model training and policy optimization. The experiment mirrors the setting described in Section 7.1. The codes and annotations for the computed winrates for ppo are now available in our anonymous github repository [https://anonymous.4open.science/r/SPO-4159].
> > >
> > > We computed the win-rate of PPO against the SFT model using AlpacaEval. The results are as follows:
> > >
> > >
> > > | Algorithm | win-rate (%)   | LC win-rate (%) |
> > > |--------------|--------|--------|
> > > | PPO       | 56.10  |  50.67 |
> > > | SPO       | **61.63**  |  **56.25** |
> > >
> > > These results will be added to Table 1 in the final version of the paper. We also plan to test PPO in our the currently running Llama3-8B experiments, whose complete results will be reported in the final version.
> > >
> > >
> > > > You can say the learned reward model may overestimate out-of-domain samples, but this problem is also encountered by reward-free policy optimization algorithms as the policy is only trained on in-domain data generated by some unknown policy, and the performance at inference time is not guaranteed.
> > >
> > > We agree with your point, and acknowledge the limitation of reward-free algorithms on small datasets overestimating out-of-domain samples. This aligns with a core contribution of our paper: SPO uses out-of-dataset regularization to address this challenge. Section 7.3 ablates the impact of this regularization, and Section 6 discusses its inspiration from RLHF methodology.
> > >
> > > > Also, PPO is not an off-policy algorithm and we only use off/on-policy to describe the policy optimization algorithm but not the reward modeling phase.
> > >
> > >
> > > We appreciate your clarification regarding on/off-policy terminology in LLM contexts. Our previous usage followed Reinforcement Learning conventions per (Sutton & Barto 2018), where off/on-policy is commonly used for both prediction (i.e., value estimation) and control. To prevent confusion, we avoid on/off-policy references in the paper.
> > >
> > >
> > > We welcome further feedback and would greatly appreciate if you consider updating your score based on these clarifications and additional results.

---

> > > > ### Comment · Reviewer_nmc9 · 2024-12-03
> > > >
> > > > Thanks for the author's response, I appreciate the author's efforts and I would like to raise my score given the positive results.

---

### Official Review · Reviewer_7Lkw · 2024-10-27

**Soundness:** 2
**Presentation:** 3
**Contribution:** 2
**Rating:** 5
**Confidence:** 4

**Summary:**

This paper presents Soft Preference Optimization (SPO), a generative model approach that aligns with human preferences without a reward model. SPO directly optimizes the model output by combining a natural loss function with a preference loss and a regularization term, covering the entire output distribution. It was theoretically proved that under the assumption of the Bradley-Terry model, SPO could converge to the Softmax distribution, and the output softness was adjusted by the algorithm parameters. Experiments on AlpacaFarm benchmark show that SPO outperforms SimPO, DPO and other alignment algorithms. The authors claim the advantage of SPO in avoiding determinism compared to DPO and RLHF.

**Strengths:**

1. This paper introduces the concept of Soft Preference Optimization (SPO) and elaborates on its theoretical foundations. The authors not only outline the goals and motivations behind SPO but also explore its distinctions from existing approaches, such as Direct Preference Optimization (DPO). The mathematical proofs presented in the paper are rigorous, particularly those of the main theorems detailed in the appendix. These proofs not only substantiate the effectiveness of the SPO method but also establish that SPO can converge to an optimal softmax distribution under specific conditions. Such rigorous mathematical derivations lay a solid foundation for the scientific validity and credibility of the SPO approach.

2. In the experimental section, SPO outperforms some current baseline alignment methods, demonstrating its effectiveness in tasks such as instruction following. Moreover, SPO significantly outperforms other baselines in its ability to focus on the best data.

**Weaknesses:**

1.The paper presents Soft Preference Optimization (SPO) as a novel method for aligning generative models with human preferences without relying on a reward model.  While the theoretical foundation is well-elaborated, the experimental section lacks a comprehensive evaluation of SPO across a broader range of pairwise alignment datasets and benchmarks.  A more extensive set of experiments is necessary to substantiate the claim that SPO can effectively focus on optimal data alignment without sacrificing performance in other areas.  It is crucial to include more experimental results and analyses to demonstrate that SPO enhances model alignment while maintaining or improving performance in related tasks.


2. The authors highlight the differences between SPO and existing approaches like Direct Preference Optimization (DPO) and Reinforcement Learning from Human Feedback (RLHF).  While it is noted that SPO introduces a superior regularization term to prevent overfitting on preference data, the paper does not sufficiently address whether this approach might come at the expense of other model capabilities.  DPO, being a reward-model-free algorithm, is already designed to balance alignment with human preferences and model performance.  Therefore, it is essential for the authors to conduct additional experiments that validate whether SPO's emphasis on diversity and alignment comes at the cost of the model's ability to perform well on a broader range of tasks.  A thorough analysis comparing SPO against DPO and other baselines in terms of overall model performance, as well as alignment, would significantly strengthen the paper's contributions and claims.

3. The experimental section of the paper, while demonstrating the effectiveness of Soft Preference Optimization (SPO), does not include an ablation study on the hyper-parameters used within the SPO framework. By including such analysis, the paper would offer a more complete picture of the method's robustness and the sensitivity to hyper-parameter tuning, which is valuable for both theoretical understanding and practical application.

**Questions:**

1. Does the performance of the SPO come at the expense of other capabilities?

2. Diversity and accuracy are inherently trade off, but why is there no discussion of this in the experimental section of the article?

---

> ### Author Response · Authors · 2024-11-28
> **Authors Response 1/2**
>
> We thank the reviewer for their constructive feedback and for highlighting areas where the paper can be improved. We have revised the manuscript accordingly and believe that addressing these comments has strengthened our work.
>
>
> > Need for more extensive experiments to substantiate claims and demonstrate that SPO enhances alignment without sacrificing performance in other tasks.
>
>
> We appreciate your suggestion to expand our experiments. Following your advice, we have conducted new experiments using the Llama 3-Base (8B) model on the UltraFeedback dataset following the same procedure in SimPO paper. This addresses the need to verify the generalizability of SPO across different instruction-tuning distributions and to demonstrate its effectiveness with a more capable open-source LLM.
>
> Although the experiments are ongoing, preliminary results indicate that SPO outperforms existing methods in this new setting. Specifically, we have evaluated both the SimPO algorithm (which had reportedly outperformed other baselines), DPO algorithm, and our SPO algorithm with in-dataset Dkl regularization. The **win rates (resp. LC win rates)** of these algorithms against GPT-4 are **15.38% (15.53%)** for DPO, **17.7% (17.33%)** for SimPO and **17.81% (18.23)%** for SPO with the standard devations of less than 1.1, showing that SPO surpasses SimPO. We anticipate even better performance for SPO with global Dkl regularization, and these experiments are currently underway. Implementing global Dkl required additional coding effort due to the need for online sampling, which has contributed to some delays. (Adapting our codebase for the Llama 3-8B model involved significant work. Our initial experiments were conducted using AlpacaFarm codebase for Llama 2 models, which was incompatible with Llama 3 models. Consequently, we switched to the SIMPO codebase, necessitating substantial code modifications.)
>
> We believe these additional experiments enhance the robustness and generalizability of our method. We will include the results in the final version of the paper, providing stronger evidence for the broad applicability of SPO.
>
>
> > Concern that SPO's emphasis on diversity and alignment might come at the expense of other model capabilities, and the need for thorough analysis comparing SPO against DPO and other baselines in terms of overall performance.  Does the performance of SPO come at the expense of other capabilities?
>
> This is a valid concern. Large diversity is not always helpful in all applications. The diversity of SPO however can be controlled via the parameter $\alpha$, which helps SPO to adapt to the diversity-needs of different tasks.
>
> The benchmarks used in our current experiments, in particular, do not encourage diversity. The sole focus on maximizing win-rates in these benchmarks could even discourage diverse solutions, potentially at the risk of mode collapse. Notably, SPO outperformed existing baselines over these diversity-unfriendly benchmarks. However, the full potential of SPO would be more evident in applications where diversity is important, such as reasoning tasks in LLMs which heavily rely on diverse exploration. In such applications (for example for finding an initial policy via alignment for reasoning through tree-of-thought algorithm), we anticipate SPO  to outperform existing baselines even more significantly.  In such settings, we anticipate larger values for the optimal $\alpha$. We differ empirical stuy of SPO in these settings to future works, due to our current limitations regarding computational resources.
>
> Note also that not every kind of diversity is useful. SPO aims to promote beneficial diversity by aligning the model's output distribution with the diversity present in the expert data and expert diversity would be more evident in larger datasets. Unfortunately, the scale of existing benchmarks are not large enough to demonstrate the full potential of SPO.
>
> > 3. Lack of an ablation study on hyper-parameters to demonstrate the method's robustness and sensitivity to hyper-parameter tuning.
>
>
> In the revised manuscript, we have added results on the sensitivity of SPO to hyper-parameter tuning, presented in Appendix D. Our experiments show that the optimal $\alpha$, and  $\beta$ parameters for SPO showed little variability across different benchmarks. We acknowledge the importance of hyper-parameter robustness and would be willing to conduct additional experiments with finer sweeps if you think it would be a valuable addition to the paper.
>
>
> Sensitivity of SPO winrates to hyperparameters in the Llama2-7B experiment:
>
> | | | |
> |--------------|--------|--------|
> | $\beta \backslash  \alpha$ | 0.01   | 0.001  |
> | 0.01         | **60.83**  | 59.82  |
> | 0.001        | 57.15  | 59.99  |

---

> ### Author Response · Authors · 2024-11-28
> **Authors Response 2/2**
>
> > Diversity and accuracy are inherently a trade-off, but why is there no discussion of this in the experimental section?
>
> Thanks for bringing this into our attention. In the rebuttal period, we computed Dkl values for the algorithms at the checkpoints where win rates were reported. Our findings indicate that Dkl is almost similar across different algorithms, with no statistically significant difference. Further addressing your concerns, a valuable addition to the paper would be a Dkl vs win-rate curve. This will however require a very fine sweep on hyperparameters and checkpoints, because desired curve would be the envelope of Dkl-winrate curves for different hyperparameters and checkpoints. Unfortunately we do not have enough resources to run such experiment at this stage, since our resources have been used up for the new LLama3-8B experiments added in the revision. We plan to try to make it into the final version of the paper.

---

> > ### Comment · Reviewer_7Lkw · 2024-11-29
> > **I would like to know more about the evaluation results**
> >
> > Thanks to the author for solving my doubts.
> >
> > I would also like to see the evaluation comparison results of SPO on the open llm leaderboard2. If SPO still plays its excellent ability, I will improve my score

---

### Official Review · Reviewer_Yexb · 2024-10-30

**Soundness:** 4
**Presentation:** 4
**Contribution:** 4
**Rating:** 8
**Confidence:** 3

**Summary:**

This paper introduces Soft Preference Optimization (SPO), a technique to fine-tune a LM to fit a dataset containing the preferred response to a given query, given some alternative, or given a ranking of possible responses. The main contribution of this paper is to model the preference probability $P_{\pi_\theta}(y_1 \succ y_2)$ as $\frac{\pi_\theta(y_1)}{\pi_\theta(y_1) + \pi_\theta(y_2)}$ and then fit the preference data using a log-likelihood loss plus a regularization term, which here is taken to be the commonly used KL divergence to the original model.

The model is further extended to include an (inverse) "temperature" parameter $\alpha$ to regulate the entropy of the resulting distribution, and a weighting function $\mu$ that is intended to down-weight preference data given on poor-quality samples (as judged by the model itself).

Also, a generalized loss to accommodate best-of-n preference data and ranked preference data is provided.

The model was evaluated on win-rates against a SFT-based competitor, comparing positively to a number of competitive baselines on pairwise data and also to DPO on best-of-n and ranked data. For the latter task, a dataset was generated which is provided in the anonymous link.

Finally, a (biased, low-variance) estimator of the KL is proposed using token-level subsequences of the samples, which I think is another contribution of this paper, although I am not 100% sure of that.

**Strengths:**

* S1: The paper proposes a simple and elegant model to fit preference data, including pairwise, best-of-n, and ranked data. The model is so simple in fact that it is surprising it had not been proposed before. (*1)
* S2: It also includes an extensive theoretical analysis of the optimal solution to the proposed loss when the preference data is sampled from Bradley-Terry or Plackett-Luce models (*2)
* S3: The presentation is very clear
* S4: The empirical evaluation is sufficiently convincing of the competitiveness of the method

(*1) In the related work, the authors do acknowledge a similar proposal in CPO by Xu et al.
(*2) I have not carefully checked the calculations of the theorems proofs.

**Weaknesses:**

None that I can think of

**Questions:**

1.
> Moreover, SPO has an advantage over DPO and RLHF in avoiding determinism. In cases where the
preference dataset is comparable to pre-training data size, regularization (DKL) becomes unnecessary,
and RLHF and DPO loss functions tend to produce deterministic models; that is they tend to return a
single high-quality response per query.

Doesn't that depend on the value of the $\beta$ parameter? See, for instance, Theorem 1 in https://arxiv.org/abs/2206.00761. You can see that as $\beta$ becomes large, the optimal distribution reverts to the original model.

---

> ### Author Response · Authors · 2024-11-28
> **Authors Response**
>
> We thank the reviewer for their positive feedback and for appreciating the value of our contributions.
>
> > 1."Moreover, SPO has an advantage over DPO and RLHF in avoiding determinism. In cases where the preference dataset is comparable to pre-training data size, regularization (DKL) becomes unnecessary, and RLHF and DPO loss functions tend to produce deterministic models; that is they tend to return a single high-quality response per query." Doesn't that depend on the value of the $\beta$ parameter? See, for instance, Theorem 1 in https://arxiv.org/abs/2206.00761. You can see that as $\beta$ becomes large, the optimal distribution reverts to the original model.
>
> You are absolutely correct, the determinism of RLHF and DPO solutions depends on the value of $\beta$. We modified the quoted statement in the revised paper to clarify that by "regularization (DKL) becomes unnecessary" we refer to the regime where $\beta\simeq 0$.
>
> It is notewothy that RLHF and DPO converge to a *deterministic model* for small $\beta$, and to the *reference model* for large $\beta$. **In neither case** do RLHF and DPO converge to the *expert model*. In contrast, SPO converges to the *expert model* for $\alpha=1$ and small $\beta$. Moreover the softness of the model is controllable via $\alpha$.

---

### Official Review · Reviewer_UurZ · 2024-11-02

**Soundness:** 1
**Presentation:** 3
**Contribution:** 3
**Rating:** 5
**Confidence:** 3

**Summary:**

This paper introduces SPO (Soft Preference Optimization), a method for aligning language models with human preferences without requiring a reward model. SPO operates by optimizing model outputs directly using a loss function that combines preference loss with regularization across the full output distribution. While SPO doesn't require reward model assumptions, the authors prove it converges to a softmax of scaled rewards under the Bradley-Terry model, with adjustable softmax temperature. The method differs from predecessors like DPO in two ways: its preference loss formulation, and its application of regularization across the entire output distribution rather than just the preference dataset.

**Strengths:**

1. The paper presents a well-structured framework for preference loss without relying on Bradley-Terry/Plackett-Luce model assumptions. The development from basic to general algorithm is logical and easy to follow.
2. The framework shows impressive versatility by handling multiple data types (pairwise, best-of-n, and ranked preference data), demonstrating a comprehensive approach to preference optimization.
3. The introduction of global regularization rather than in-dataset regularization is a novel and well-justified approach that shows better empirical results.
4. The theoretical foundations are solid, with clear proofs and convergence guarantees for each variant of the algorithm.

**Weaknesses:**

1. The experimental evaluation has several limitations:
* The experimental evaluation would benefit from broader model and dataset coverage. While the current results on Llama-2-7B and AlpacaFarm are present, only one setting might be not enough. Here are some suggestions for reference:
The baseline performance is concerning, with several methods showing negative results on length-controlled win rates, suggesting potential issues with the experimental setup or hyperparameter tuning. You may want to consider including models from different families such as Mistral-7B to show architecture-agnostic performance. To mention but not significant problem, llama-2-7B is currently not among the most capable open-source LLMs according to leaderboards like AlpacaEval. Showing results for llama-3-8B might make the results more significant. Additionally, evaluating on diverse datasets like UltraFeedback (~3 times larger than the alpacaFarm dataset) would verify generalizability across different instruction-tuning distributions.  These additions would provide stronger evidence for the method's broad applicability.

* Specific concerns are raised about the baseline performance that warrant further investigation. The baseline methods has at most 51.94% on the LC Win-rate against SFT, which seems unusually poor given their reported performance in original papers. Since you are fine-tuning starting from the SFT model, that means all the preference optimization baseline methods nearly not improve the quality at all, even worse the performance for R-DPO, CPO, SimPO, IPO.
                                           Is there any specific reason? While there could be various reasons, one possibility is suboptimal hyperparameter choices. I recommend using similar hyperparameter settings as successful implementations in previous work (e.g., Meng et al. 2024, SimPO paper reported strong performance with their specific configuration), to ensure a fair comparison of these baseline methods.

2. Could you clarify the evaluation dataset sizes for each experiment? This information is particularly relevant when examining the global regularizer results, where some improvements show modest gains (e.g., from 59.9% to 60.8%). Given these subtle differences, I recommend including statistical significance testing to strengthen the results' interpretation.

**Questions:**

1. Regarding the preference representation: While your method defines $p(y_w > y_l | x)$ without depending on BT/PL model assumptions, isn't it effectively equivalent in expressivity? Please correct me if I am wrong. It seems that $\pi_\theta(y_w | x)^\alpha$ could be seen as an alternative parametrization of $\exp(\frac{1}{\beta}r_\phi(y_w, x))$, where $\alpha$ and $\beta$ serve similar roles, and $r_\phi$ and $\pi_\theta$ map to equivalent functions. Could you clarify if/how your formulation provides additional expressivity?
2. How does the choice of $\alpha$ interact with different types of preference data (pairwise, best-of-n, ranked)? Is there a principled way to select optimal $\alpha$ values for different scenarios or data distributions?

---

> ### Author Response · Authors · 2024-11-28
> **Authors Response 1/2**
>
> We thank the reviewer for their feedback and for raising practical questions, which greatly helped to improve the paper. We have revised the manuscript accordingly and believe that addressing these comments has strengthened the solidity of the paper and the presented results.
>
>
> > 1. The experimental evaluation has several limitations: The experimental evaluation would benefit from broader model and dataset coverage. While the current results on Llama-2-7B and AlpacaFarm are present, only one setting might be not enough. You may want to consider including models from different families such as Mistral-7B to show architecture-agnostic performance. To mention but not significant problem, llama-2-7B is currently not among the most capable open-source LLMs according to leaderboards like AlpacaEval. Showing results for llama-3-8B might make the results more significant. Additionally, evaluating on diverse datasets like UltraFeedback (~3 times larger than the alpacaFarm dataset) would verify generalizability across different instruction-tuning distributions. These additions would provide stronger evidence for the method's broad applicability.
>
> Following your advice, we have conducted new experiments to broaden our model and dataset coverage. Specifically, we performed experiments using the Llama3-8B model on the UltraFeedback dataset follwoing the exact procedure in SimPO paper. This addresses your suggestion to verify generalizability across different instruction-tuning distributions and to provide more significant results with a more capable open-source LLM.
>
> Although the experiments are still ongoing, we have obtained preliminary results. So far, we have evaluated the SimPO algorithm (which has reportedly outperformed previous baselines), DPO algorithm, and  SPO algorithm with in-dataset Dkl regularization. The **win rates (resp. LC win rates)** of these algorithms against GPT-4 are **15.38% (15.53%)** for DPO, **17.7% (17.33%)** for SimPO and **17.81% (18.23)%** for SPO with the standard devations of less than 1.1, indicating that SPO method outperforms SimPO even in this new setting.
>
> We anticipate even better performance for SPO with global Dkl regularization, and these experiments are currently underway. Implementing global KL divergence required additional coding effort due to the need for online sampling, which has contributed to some delays.
>
> Adapting our codebase for the Llama 3-8B model involved significant work. Our initial experiments were conducted using AlpacaFarm codebase for Llama 2 models, which appeared to be incompatible with Llama 3 models. Consequently, we switched to the SIMPO codebase, necessitating substantial code modifications. This transition was a major reason for the delay in submitting our responses.
>
> We believe that these additional experiments significantly enhance the robustness and generalizability of our method. We will include the results in the final version of the paper, providing stronger evidence for the broad applicability of our approach.
>
>
>
>
> > The baseline performance is concerning, with several methods showing poor results on length-controlled win rates, which suggests potential issues with the experimental setup or hyperparameter tuning. The highest LC win rate against the SFT model is only 51.94%, unusually low compared to their reported performance in original papers. Since you're fine-tuning from the SFT model, it appears that all the preference optimization baselines do not improve quality and may even worsen it (e.g., R-DPO, CPO, SimPO, IPO). Is there a specific reason for this? One possibility could be suboptimal hyperparameter choices. I recommend using similar hyperparameter settings as successful implementations in previous work (e.g., Meng et al., 2024; SimPO reported strong performance with their specific configuration) to ensure a fair comparison.
>
>
> The reason is that we tuned hyperparamters for win-rates only, and reported both win-rates and length-controlled win-rates on the same checkpoints. Notabaly, SPO shows better generalization to length-controlled win rates compared to other baselines. In the revised manuscript, we have noted your observation in the experiments section and clarified that the length-controlled win rates were evaluated on models optimized for win rates. (This is the case both for Llama2 experiments in the paper and Llama3 experiments mentioned in the previous comment).
>
> We are confident in our hyperparameter sweeps, as we thoroughly explored each parameter over the ranges specified in the original papers. Details of the hyperparameters and sweep ranges are provided in Appendix D1.
>
> Due to resource constraints, we were unable to conduct a separate hyperparameter sweep for length-controlled win rates at this time, as our efforts were focused on the Llama 3-8B experiments. However, we would be happy to perform this additional sweep for the camera-ready version if you believe it would be a valuable addition.

---

> ### Author Response · Authors · 2024-11-28
> **Authors Response 2/2**
>
> > 2. Could you clarify the evaluation dataset sizes for each experiment? This information is particularly relevant when examining the global regularizer results, where some improvements show modest gains (e.g., from 59.9% to 60.8%). Given these subtle differences, I recommend including statistical significance testing to strengthen the results' interpretation.
>
> Thank you for bringing this valid point to our attention. The sizes of the evaluation datasets used in our experiments and the resulting standard deviation in win-rates are as follows:
>
> - AlpacaEval: 805 samples, resulting in a standard deviation of less than 1.5% (as mentioned in Appendix D.2).
> - TinyStories (Best-of-n): 1,000 sample pairs, with a standard deviation of about 1.4%.
> - TinyStories (Ranking): 1,000 sample pairs, also yielding a standard deviation of approximately 1.4%.
>
> As you noted, the statistical significance testing is important, especially when improvements are modest. This is a common issue in the alignment literature, where most works report improvements of only 1–2% over baselines. Unfortunately, expanding dataset sizes to reduce variance is not always feasible, particularly for datasets like AlpacaEval where data availability is limited.
>
> Please be assured that we did not cherry-pick seeds or results; the win rates reported are sample averages that represent unbiased estimate of the true win-rates. To enhance the interpretation of our findings, we have now included confidence intervals in the experiments section of the revised manuscript.
>
>
> > Question 1. Regarding the preference representation: While your method defines $p(y_w>y_l|x)$ without depending on BT/PL model assumptions, isn't it effectively equivalent in expressivity? Please correct me if I am wrong. It seems that $\pi_{\theta}(y_w|x)^{\alpha}$ could be seen as an alternative parametrization of $\exp(\frac{1}{\beta}r_{\phi}(y_w|x))$, where $r_{\phi}$ and $\pi_{\theta} serve similar roles, and  and map to equivalent functions. Could you clarify if/how your formulation provides additional expressivity?
>
> This is an insightful observation to note the connection between the loss function used to train the reward model in RLHF and the preference-loss in SPO.  In fact, $\pi_{\theta}(y_w|x)^{\alpha} = \exp(\frac{1}{\beta}r_{\phi}(y_w|x))$ if $r_{\phi}(y_w|x)=\log \pi_{\theta}(y_w|x)$ and $\alpha=1/\beta$. Despite this similarity, the roles and expressivity of $\pi_\theta$ and $r_\phi$ differ significantly, because $\pi_\theta$ is a generative model while the reward-model $r_{\phi}$ is a value model with scalar outputs. Note that generating high-quality responses is typically more challenging than evaluating the quality of a given response, a fact that has been leveraged in recent progress in LLM reasoning, such as tree-of-thought planning. In summary, the generative model $\pi_\theta$ is generally more expressive than the reward model $r_\phi$.
>
> The difference between RLHF and SPO resembles the difference between actor-critic methods and policy-gradient methods  in RL. Actor-critic methods (here RLHF)  train a value model (here reward model) and use it in the updating the policy network (here the generative LLM), whereas policy-gradient directly updates the policy network based on the observations without a need for value models. In fact, in retrospect, it is quite surprising that RLHF was chronologically developed before SPO, given the simpler pipeline of SPO which also avoids bias transfer from reward models.
>
>
> > Question 2. How does the choice of $\alpha$ interact with different types of preference data (pairwise, best-of-n, ranked)? Is there a principled way to select optimal $\alpha$ values for different scenarios or data distributions?
>
>
> The choice of $\alpha$ primarily influences the "softness" or entropy of the model's output distribution. In applications that benefit from exploration and diversity—such as LLM reasoning—a larger $\alpha$ is preferred because it promotes softer and more diverse policies, as indicated by Theorems 1–4 in our paper. We anticipate that SPO will outperform other methods even more significantly in such domains.
>
> Existing alignment benchmarks focus on maximizing win rates without encouraging diversity, and without discouraging mode collapse that is more likely in less diverse methods. Therefore, our current experiments may not fully showcase SPO's potential in promoting diversity.
>
> We also expect the optimal value of $\alpha$ to increase with larger and more diverse preference datasets. A larger $\alpha$ allows the model to capture more nuanced preferences present in expansive datasets. In our parameter sweeps reported in Appendix D1, we found that the optimal $\alpha$ for SPO was consistent (equal to 0.01) across different types of preference data, including pairwise, best-of-$n$, and ranked preferences.
>
> We added a discussion of the above points to Section 9 of the revised paper.

---

> ### Author Response · Authors · 2024-12-03
>
> We sincerely appreciate the time and effort you have devoted to reviewing our work and for sharing your valuable insights. We have carefully addressed the concerns and suggestions you raised and have provided detailed responses to each point.
>
> As the open discussion phase is nearing its conclusion, we would greatly appreciate any final feedback or thoughts you might have on our revisions and in light of our responses.
>
> Thank you again for your time and contributions to this process.

---

### Official Review · Reviewer_4yoQ · 2024-11-02

**Soundness:** 2
**Presentation:** 3
**Contribution:** 3
**Rating:** 5
**Confidence:** 3

**Summary:**

This paper proposes an offline alignment objective composed of the preference loss and regularization loss. In the preference loss, it uses a temperature parameter to control the "softness" of the preference; in the regularization loss, it uses KL under different prompts (than preference prompts). Theoretically, it shows the connection to the reward models. Empirically, it shows improved win rates against SFT among baselines

**Strengths:**

* The paper studies the important question of what regularization data for the alignment methods
* The proposed objective is interesting and has connection to the reward model

**Weaknesses:**

I think the main weakness is that the proposed method is a combination of many components, and it's difficult to assess the utility of each one
* First, the preference loss rewritten as $log \sigma(log \pi_\theta(y_1) - log \pi_\theta(y_2))$ is similar to the DPO loss, but without calibration using $\pi_0(.)$. The DPO has the calibration terms because of the KL regularization using in-distribution dataset. The paper can't convince me that getting rid of the calibration terms brings any advantage in itself (either theoretically or empirically). I think the calibration term could be important when for example $y_1$ is much more/less likely than $y_2$ under $\pi_0$. Maybe the author could consider an ablation experiment comparing the two objectives (you can also use $\alpha$ in both cases too).
* Second question is when might out-of-domain regularization helps. Again, I am not convinced if, and when, such regularization helps.

Besides,
* I can't confidently assess the method's utility just looking at win rates in the experiments. A method could do very well in win rate by having a huge KL from the SFT --- maybe plotting KL vs win rate gives more convincing arguments.
* The paper says RLHF uses out-of-domain prompts --- is that true?

I will raise my points if my concerns get addressed.

**Questions:**

I put my questions in the weakness

---

> ### Author Response · Authors · 2024-11-28
> **Authors Response 1/3**
>
> We thank the reviewer for their feedback and for raising deep questions about the utility of the core ideas proposed in the paper. These insights indicate that our initial presentation may not have effectively conveyed the key concepts. Accordingly, we have revised the paper to enhance clarity and address the concerns raised.
>
> > First, the preference loss rewritten as $\log(\sigma(\log(\pi(y_1))-\log(\pi(y_2))))$ is similar to the DPO loss, but without calibration using $\pi_0(.)$. The DPO has the calibration terms because of the KL regularization using in-distribution dataset. The paper can't convince me that getting rid of the calibration terms brings any advantage in itself (either theoretically or empirically). I think the calibration term could be important when for example $y_1$ is much more/less likely than $y_2$ under $\pi_0(.)$. Maybe the author could consider an ablation experiment comparing the two objectives (you can also use $\alpha$ in both cases too).
>
> Eliminating the calibration terms enables the model to converge to a truly aligned model, rather than a partially aligned one. For a concrete illustration of this advantage, let's focus on the scenario you mentioned. Suppose the preferred response $y_w$ is much less likely than the less preferred response $y_l$ under the reference model $\pi_{\text{ref}}$, so that $\frac{\pi_{\text{ref}}(y_w)}{\pi_{\text{ref}}(y_l)} = \epsilon \ll 1.$
>
> Since the expert strongly prefers $y_w$ over $y_l$, an ideally aligned model $\pi_\theta$ should satisfy $\frac{\pi_\theta(y_w)}{\pi_\theta(y_l)} > 1$. However, the calibration term in DPO can only enforce $\frac{\pi_\theta(y_w)}{\pi_\theta(y_l)} > \frac{\pi_{\text{ref}}(y_w)}{\pi_{\text{ref}}(y_l)} = \epsilon,$ which is significantly less than 1. This means that even after optimization, the DPO model may still prefer $y_l$ over $y_w$, failing to achieve proper alignment.
>
> In contrast, the SPO loss, not having the calibration term, allows the model to fully align with the expert preferences by ensuring $\frac{\pi_\theta(y_w)}{\pi_\theta(y_l)} > 1$.  Thus, eliminating the calibration terms enables the model to converge to a truly aligned state, rather than a partially aligned one as in DPO.
>
> The calibration term in DPO is a kind of regularization aiming to improve generalization, however (as discussed in the next comment) this approach is not the most effective strategy for ensuring generalization.

---

> ### Author Response · Authors · 2024-11-28
> **Authors Response 2/3**
>
> > Second question is when might out-of-domain regularization helps. Again, I am not convinced if, and when, such regularization helps.
>
>
> **Clarification of terminology:**
>
> There might be a nuanced difference between **out-of-domain** regualization and **out-of-dataset** regualrization (which we also call global regularization).
> While out-of-domain data may involve query-response pairs whose *query* is not in the preference dataset, by out-of-dataset data we refer to query-response pairs whose *response* is not in the preference dataset. In the latter case, the query may or may not be in the dataset, but in either case should be relevant to the test domain. In our experiments, we simply sampled queries from those available in the dataset because we did not assume access to a broader test domain. Note that the paper motivates  out-of-dataset regualirzation, not out-of-domain regualrization. We explicitely mentioned this distinction in the Section 6 of the revised paper.
>
>
>
> **Benefit of global regularization:**
>
> Global regularization is beneficial when the preference dataset is relatively small compared to the dataset used to train the reference model $\pi_{\text{ref}}$, which is typically the case in alignment tasks. Preference datasets often consist of only a few thousand tokens, whereas pre-training datasets can contain trillions of tokens. This disparity can lead to unintended out-of-dataset distribution shifts when optimizing over the relatively tiny preference dataset alone.
>
> In RLHF, Dkl regularization aims to prevent large distribution shifts in $\pi_\theta$ for data points not present in the preference dataset. Regularizing only over the preference dataset is insufficient to mitigate these shifts in unseen data. Large distribution shifts are problematic outside the preference dataset because we lack labeled preferences for those instances. Within the preference dataset, we intentionally shift the distribution to align with the provided labels. Therefore, applying a global KL regularizer better serves the goal of maintaining overall model behaviour while aligning with preference data.
>
> **A toy exmaple (failiure of DPO):**
>
> To illustrate the aobve points, consider a simplified example where there is a single question, $x$, and with four possible responses: $y_1, y_2, y_3, y_4$. Let the reference model $\pi_{\text{ref}}$ be uniform, so $\pi_{\text{ref}}(y_i | x) = 0.25$ for $i=1,2,3,4$. Suppose our preference dataset $D‎ = \{( y_2 \succ y_1| x )\}$ contains only one sample indicating a preference: $y_2 \succ y_1$ given $x$.
>
> Assume that the model $\pi_\theta$ is parameterized such that $\pi_\theta(y_2 | x)=0.5-\pi_\theta(y_1 | x)$, $\pi_\theta(y_4 | x)=0.5-\pi_\theta(y_3 | x)$,  $\pi_\theta(y_1 | x) = \sigma(\theta)/2$ and $\pi_\theta(y_3 | x) = \sigma(c\theta)/2$, where $\sigma$ is the sigmoid function and  $c$ is a fixed constant (not a learnable weight). The DPO loss always ignores out-of-dataset query-response pairs, and in particular ignores $\pi_\theta(y_3|x)$ and $\pi_\theta(y_4|x)$ in this scenario. Consequently DPO loss is unaffected by the value of $c$; thus, it converges to the same $\theta$ regardless of $c$'s magnitude. This is problematic because if $c$ is large, the model experiences a significant distribution shift for $y_3$ and $y_4$, for which we have no preference data.
>
>
> **SPO's global regularization resolves the problem in the above example:**
>
> Under a global regularizer, as in the SPO method, the model adjusts $\theta$ more cautiously when $c$ is large, preventing unintended shifts in $\pi_\theta(y | x_2)$. This behaviour ensures that the model doesn't overfit to the small preference dataset at the expense of overall performance. In real-world applications with larger models, the number of undesirable responses vastly outnumbers the desirable ones. Therefore, unregulated distribution shifts are more likely to increase the probability of generating poor responses.
>
> **In summary:**
>
> For the model to generalize well outside the preference dataset, applying regularization globally is more effective. We have conducted an ablation study comparing global and in-dataset regularization, with results presented in Section 7.3. The results show that global regularization leads to better models compared to in-dataset regularization. In the revised manuscript, we disucss the above advantages in Section 6.

---

> ### Author Response · Authors · 2024-11-28
> **Authors Response 3/3**
>
> > The proposed method is a combination of many components, and it's difficult to assess the utility of each one.
>
> SPO has two main components: preference loss and regularization. We conducted an ablation study on the impact of global vs in-dataset regularization on the performance, and report in Section 7.3. Regarding the preference loss, finding a suitable alternative to combine with a global regularizer for ablation studies is challenging. We have reported the overall performance of different algorithms using their specific regularizers. We welcome any specific suggestions the reviewer may have for further ablation studies in this area.
>
> We acknowledge that the current experiments in the paper do not capture the full utility and potentials of SPO, even though SPO has outperformed other alignment methods in these experiments. In particular, we anticipate larger performance gaps between SPO and baselines in the following case: 1- in applications that greatly benefit from diversity and softness of the policy (like exploration in LLM reasoning tasks; as opposed to mere win-rate maximization benhmarks), 2- upon availability of more compute to acquire better approximation of global Dkl (as opposed to our current implementation that uses intermittent sampling to reduce the cost of online Dkl computation), and 3- over larger scale datasets which would help SPO to obtain better approximation of the expert policy (as backed by the presented theoretical results in the paper). Unfortunately we could now carry out these experiemnts due to our resource constraints at this time, and therefore had to focus on standard win-rate maximization benchmarks. We believe the above tests are promising avenues to better showcase the utility of SPO in future works.
>
>
> > I can't confidently assess the method's utility just looking at win rates in the experiments. A method could do very well in win rate by having a huge KL from the SFT --- maybe plotting KL vs win rate gives more convincing arguments.
>
>
> We agree that relying solely on win rates may not fully capture the utility of the method. To address this concern in the revision, we computed Dkl values for the algorithms at the checkpoints where win rates were reported. Our findings indicate that the differences between Dkl across different algorithms at corresponding checkpoints are not statistically significant, and all have the same order.
>
> We acknowledge that plotting Dkl versus win rate would provide a more comprehensive evaluation. However, generating such trade-off curves fairly for different algorithms would require an extensive sweep over hyperparameters and checkpoints for deriving the envelope of Dkl-winrate curves for different hyperparameters and checkpoints for each method. Due to computational constraints—particularly because our resources were allocated to the new LLama3-8B experiments—we were unable to conduct these additional experiments at this time. We plan to include these plots in the final version of the paper.
>
>
> We also agree that win-rates do not necessarily reflect the utility of a loss function, since achieved win-rates rely not only on the loss function but also on the optimization algorithm used to optimize it. We believe that the SPO loss reflects the true objective of alignment, that is aligning to expert preferences while preventing unwanted distribution shifts.
>
>
>
> > The paper says RLHF uses out-of-domain prompts --- is that true?
>
> RLHF uses out-of-dataset responses rather than out-of-domain prompts. After learning a reward-model, in the policy optimization phase of RLHF the Dkl loss is used to regularize the divergence from $\pi_{ref}$. This policy optimization phase often involves online sampling of responses from $\pi_\theta$, akin to the Dkl computation in SPO. Therefore the responses used in RLHF are not limited to the preference dataset.

---

> ### Author Response · Authors · 2024-12-03
>
> We sincerely appreciate the time and effort you have devoted to reviewing our work and for sharing your valuable insights. We have carefully addressed the concerns and suggestions you raised and have provided detailed responses to each point.
>
> As the open discussion phase is nearing its conclusion, we would greatly appreciate any final feedback or thoughts you might have on our revisions and in light of our responses.
>
> Thank you again for your time and contributions to this process.

---

### Author Response · Authors · 2024-11-28
**General Response**

We thank all the reviewers for their constructive feedback, which has significantly helped us improve our paper. In response to the concerns about the breadth and robustness of our experimental evaluation, we have conducted new experiments using the Llama3-8B model on the UltraFeedback dataset, as suggested. We have also addressed the reviewers' concerns regarding the impact of SPO on other model capabilities, the role of regularization, hyperparameter sensitivity, and theoretical foundations. The revision has enhance the presentation and rigour of the paper and demonstrate that SPO improves alignment with human preferences without unintended trade-offs.

---

### Meta-Review · Area_Chair_ez1m · 2024-12-19

**Metareview:**

This paper proposes SPO (Soft Preference Optimization), a method for aligning language models with human preferences without requiring a reward model. After rebuttal, it received mixed scores of 55568. On one hand, reviewers commented that the proposed objective is interesting and insightful. The reviewer who gave a score of 8 also commented that none of the pointed out weaknesses seriously undermine the soundness or the novelty of this work. On the other hand, several reviewers also commented that the experiment evaluation is not sufficient enough to show the effectiveness of the proposed method. The experimental evaluation would benefit from broader model and dataset coverage. During rebuttal, the authors have made clarifications and added new experiments. Overall, the flaws slightly outweigh the merits, and the AC would have to make the hard decision to reject the paper by the end.

**Additional Comments On Reviewer Discussion:**

The authors have done an actually good job of rebuttal, with 2 reviewers increasing their scores.

1. Reviewers have requested additional experiments using different LLMs. During rebuttal, the authors have added results of Llama-3-8B with the UltraFeedback dataset.

2. Reviewers have requested the comparison with PPO. During rebuttal, the authors have added results of using PPO under the original Lllma-2-8B model, and showed SPO achieved better performance than PPO.

3. Reviewers have questioned about the derivation of Theorem 1, where the regularization term is not accounted for. During rebuttal, the authors provided proper responses but also admitted the potential limitation of the proof.

4. Reviewers have asked additional ablation study regarding the hyper-parameters used within the SPO framework. The authors have added some ablations on this.

5. Reviewers further asked the evaluation comparison results of SPO on the open llm leaderboard2; however, the authors did not have enough time to get this done.

---

### Decision · Program_Chairs · 2025-01-22

Reject